# Strategic Linear Contextual Bandits

**Thomas Kleine Buening**
The Alan Turing Institute

**Aadirupa Saha**
Apple ML Research

**Christos Dimitrakakis**
University of Neuchatel

**Haifeng Xu**
University of Chicago

## Abstract

Motivated by the phenomenon of strategic agents gaming a recommender system to maximize the number of times they are recommended to users, we study a strategic variant of the linear contextual bandit problem, where the arms can strategically misreport privately observed contexts to the learner. We treat the algorithm design problem as one of *mechanism design* under uncertainty and propose the Optimistic Grim Trigger Mechanism (OptGTM) that incentivizes the agents (i.e., arms) to report their contexts truthfully while simultaneously minimizing regret. We also show that failing to account for the strategic nature of the agents results in linear regret. However, a trade-off between mechanism design and regret minimization appears to be unavoidable. More broadly, this work aims to provide insight into the intersection of online learning and mechanism design.

## 1 Introduction

Recommendation algorithms that select the most relevant item for sequentially arriving users or queries have become vital for navigating the internet and its many online platforms. However, recommender systems are susceptible to manipulation by strategic agents seeking to artificially increase their frequency of recommendation [31, 33, 38]. These agents, ranging from sellers on platforms like Amazon to websites aiming for higher visibility in search results, employ tactics such as altering product attributes or engage in aggressive search engine optimization [29, 32]. By gaming the algorithms, agents attempt to appear more relevant than they actually are, often compromising the integrity and intended functionality of the recommender system. Here, the key issue lies in the agents' incentive to manipulate the learning algorithm to maximize their utility (i.e., profit). To address this challenge, we study and design algorithms in a strategic variant of the linear contextual bandit, where the agents (i.e., arms) have the ability to misreport privately observed contexts to the learner. Our main contribution is connecting online learning with approximate mechanism design to minimize regret while, at the same time, discouraging the arms from gaming our learning algorithm.

The contextual bandit [2, 24] is a generalization of the multi-armed bandit problem to the case where the learner observes relevant contextual information before pulling an arm. It has found application in various domains including healthcare [37] and online recommendation [25]. We here focus on the linearly realizable setting [1, 6], where each arm's reward is a linear function of the arm's context in the given round. In the standard linear contextual bandit, at the beginning of round $t$, the learner observes the context $x_{t,i}^* \in \mathbb{R}^d$ of every arm $i \in [K]$, selects an arm $i_t$, and receives a reward drawn from a distribution with mean $\langle \theta^*, x_{t,i_t}^* \rangle$ where $\theta^* \in \mathbb{R}^d$ is an unknown parameter. In the *strategic linear contextual bandit*, we assume that each arm is a self-interested agent that wants to maximize the number of times it gets pulled by manipulating its contexts.

More precisely, we consider the situation where each arm $i$ *privately* observes its true context $x_{t,i}^*$ every round, e.g., its relevance to the user arriving in round $t$, but reports a potentially gamed context vector $x_{t,i}$ to the learner. The learner does not observe the true contexts, but only the reported contexts

38th Conference on Neural Information Processing Systems (NeurIPS 2024).

$\mathcal{X}_t = \{x_{t,1}, \ldots, x_{t,K}\}$ and chooses an action from this gamed action set $\mathcal{X}_t$. When the learner pulls arm $i_t$, the learner then observes a reward $r_{t,i_t}$ drawn from a distribution with mean $\langle \theta^*, x^*_{t,i_t} \rangle$. In other words, the arms can manipulate the contexts the learner observes, but cannot influence the underlying reward. This is often the case as superficially changing attributes or meta data has no effect on an item's true relevance to a user.

In summary, our contributions are:

- We introduce a strategic variant of the linear contextual bandit problem, where each arm, in every round, can misreport its context to the learner to maximize its utility, defined as the total number of times the learner selects the arm over $T$ rounds (Section 3). We demonstrate that incentive-unaware algorithms, which do not explicitly consider the incentives they (implicitly) create for the arms, suffer linear regret in this strategic setting when the arms respond in Nash Equilibrium (NE) (Proposition 3.3). This highlights the necessity of integrating mechanism design with online learning techniques to minimize regret in the presence of strategic arms.

- We begin with the case where $\theta^*$ is known to the learner in advance (Section 4). This simplifies the problem setup, allowing us to establish fundamental concepts while highlighting the challenges of designing a sequential mechanism *without* payments. For this scenario, we propose the Greedy Grim Trigger Mechanism (GGTM), which incentivizes the arms to be approximately truthful while minimizing regret. We show that **(a)** Truthful reporting is an $\tilde{\mathcal{O}}(\sqrt{T})$-NE for the arms (Theorem 4.1) and **(b)** GGTM has $\tilde{\mathcal{O}}(K^2\sqrt{KT})$ regret under *every* NE of the arms (Theorem 4.2).

- Next, we consider the case where $\theta^*$ is unknown to the learner in advance (Section 5). Without access to the true contexts, estimating $\theta^*$ accurately appears intractable, as the arms can manipulate the estimation process. Surprisingly, we show that learning $\theta^*$ is not necessary for minimizing regret in the strategic linear contextual bandit. We construct confidence sets (which may not contain $\theta^*$) to derive pessimistic and optimistic estimates of our expected reward. These estimates are used to construct the Optimistic Grim Trigger Mechanism (OptGTM). Despite possibly incorrect estimates of $\theta^*$, OptGTM bounds the impact of misreported contexts on both regret *and* the utility of all arms. We show that **(a)** Truthfulness is an $\tilde{\mathcal{O}}(d\sqrt{KT})$-NE for which OptGTM has regret $\tilde{\mathcal{O}}(d\sqrt{KT})$ (Theorem 5.1) and **(b)** OptGTM incurs at most $\tilde{\mathcal{O}}(dK^2\sqrt{KT})$ regret under *every* NE of the arms (Theorem 5.2).

- Finally, we support our theoretical findings with simulations of strategic gaming behavior in response to OptGTM and LinUCB (Section 6). We simulate how strategic arms adapt what contexts to report over time by equipping the arms with decentralized gradient ascent and letting the arms (e.g., vendors) and the learner (e.g., platform) repeatedly interact over several epochs. The experiments confirm the effectiveness of OptGTM and illustrate the shortcomings of incentive-unaware algorithms, such as LinUCB.

## 2    Related Work

**Linear Contextual Bandits.** In related work on linear contextual bandits with adversarial *reward* corruptions [3, 20, 39, 45], an adversary corrupts the reward observation in round $t$ by some amount $c_t$ but not the observed contexts. In this problem, the optimal regret is given by $\Theta(d\sqrt{T} + dC)$, where $C := \sum_t |c_t|$ is the adversary's budget. To the best of our knowledge, adversarial *context* corruptions have only been studied by [8], who achieve $\tilde{\mathcal{O}}(d\tilde{C}\sqrt{T})$ regret with $\tilde{C} := \sum_{t,i} \|x^*_{t,i} - x_{t,i}\|$, where $x^*_{t,i}$ and $x_{t,i}$ are the true and corrupted contexts, respectively. In contrast, we do not assume a bounded corruption budget so that these regret guarantees become vacuous (cf. Proposition 3.3). Moreover, instead of taking the worst-case perspective of purely adversarial manipulation, we assume that each arm is a self-interested agent maximizing their own utility.

**Strategic Multi-Armed Bandits.** Braverman et al. [4] were the first to study a strategic variant of the multi-armed bandit problem and considered the case where the pulled arm privately receives the reward and shares only a fraction of it with the learner. An extension of this setting has recently been studied in [12]. In other lines of work, [9, 13] study the robustness of bandit learning against strategic manipulation, however, simply assume a bounded manipulation budget instead of performing mechanism design. [11, 35] consider multi-armed bandits with replicas where strategic agents can submit replicas of the same arm to increase the number of times one of their arms is pulled. Buening et al. [5] combine multi-armed bandits with mechanism design to discourage clickbait in online

---

**Interaction Protocol 1:** Strategic Linear Contextual Bandits

---

**1** Learner publicly commits to algorithm $M$

**2** **for** $t = 1, \ldots, T$ **do**

**3** $\quad$ Every arm $i \in [K]$ privately observes its context $x_{t,i}^* \in \mathbb{R}^d$

**4** $\quad$ Every arm $i \in [K]$ reports a (potentially gamed) context $x_{t,i} \in \mathbb{R}^d$ to the learner

**5** $\quad$ Learner observes the gamed contexts $\mathcal{X}_t = \{x_{t,1}, \ldots, x_{t,K}\}$, selects arm $i_t \in [K]$, and receives reward

$$r_{t,i_t} := \langle \theta^*, x_{t,i_t}^* \rangle + \eta_t,$$

$\quad$ where $\eta_t$ is zero-mean sub-Gaussian noise. Note that the reward is generated with respect to the unknown parameter $\theta^* \in \mathbb{R}^d$ and the unobserved true context $x_{t,i_t}^*$.

---

recommendation. In their model, each arm maximizes its total number of clicks and is characterized by a strategically chosen click-rate and a fixed post-click reward. However, all of these works substantially differ from our work in problem setup and/or methodology.

**Modeling Incentives in Recommender Systems.** A complementary line of work studies content creator incentives in recommender systems [16, 21, 22, 23, 27, 40, 41, 42] and how algorithms shape the behavior of agents more generally [7]. These works primarily focus on modeling content creator behavior and studying content creator incentives under existing algorithms. Instead, our goal is the design of incentive-aware learning algorithms which incentivize content creators (arms) to act in a desirable fashion (truthfully) while maximizing the recommender system's performance.

**Strategic Learning.** We also want to mention the extensive literature on strategic learning [14, 15, 18, 19, 26, 43, 44] and strategic classification [10, 17, 34, 36]. Similarly to the model we study in this paper, the premise is that rational agents strategically respond to the learner's algorithm (e.g., classifier) to obtain a desired outcome. However, the learner interacts with the agents only once and the agents are assumed to be myopic and to suffer a cost for, e.g., altering their features. Moreover, there is no competition among the agents like in the strategic linear contextual bandit. In contrast to these works, we wish to design a sequential (online learning) mechanism to incentivize truthfulness, which is only possible because we repeatedly interact with the same set of agents (i.e., arms).

## 3 Strategic Linear Contextual Bandits

We study a strategic-variant of the linear contextual bandit problem, where $K$ strategic agents (i.e., arms) aim to maximize their number of pulls by misreporting privately observed contexts to the learner. The learner follows the usual objective of minimizing regret, i.e., maximizing cumulative rewards, despite not observing the true contexts. We here focus on the case where the strategic arms respond in *Nash equilibrium* to the learning algorithm. The interaction between the environment, the learning algorithm, and the arms is specified in Interaction Protocol 1.

Notice that the arms can manipulate the contexts that the learner observes (and the learner only observes these gamed contexts and never the actual contexts), but the rewards are generated w.r.t. the true contexts. Moreover, if all arms are non-strategic and—irrespective of the learning algorithm—report their features truthfully every round, i.e., $x_{t,i} = x_{t,i}^*$ for all $(t,i) \in [T] \times [K]$, the problem reduces to the standard non-strategic linear contextual bandit.

### 3.1 Strategic Arms and Nash Equilibrium

We assume that each arm $i$ reports a possibly gamed context $x_{t,i}$ to the learner after observing its true context $x_{t,i}^*$ and potentially other information. For example, the arms may have prior knowledge of $\theta^*$ and observe the identity of the selected arm at the end of each round. However, we do not use any specific assumptions about the observational model of the arms. Our results can be viewed as a worst-case analysis over all such models. For concreteness, consider the case where the arms have prior knowledge of $\theta^*$ and, at the end of every round, observe which arm was selected and the generated reward.[1]

---

[1]We naturally expect that the more information the arms observe each round, the more challenging the problem becomes for the learner, as the arms' ability to manipulate the learning algorithm increases.

Let $\sigma_i$ be a (mixed) strategy of arm $i$ that is history-dependent and in every round $t$ maps from observed true contexts $x_{t,i}^*$ to a distribution over reported contexts $x_{t,i}$ in $\mathbb{R}^d$. We define $\sigma_{-i}$ as the strategies of all arms except $i$ and define a strategy profile of the arms as $\boldsymbol{\sigma} := (\sigma_1, \dots, \sigma_K)$. We call arm $i$ *truthful* if it truthfully reports its privately context every round, i.e., $x_{t,i} = x_{t,i}^*$ for all $t \in [T]$. This truthful strategy is denoted $\sigma_i^*$ and we let $\boldsymbol{\sigma}^* = (\sigma_1^*, \dots, \sigma_K^*)$.

We now formally define the objective of the arms. Let $n_T(i) := \sum_{t=1}^T \mathbb{1}\{i_t = i\}$ be the number of times arm $i$ is pulled by the learner's algorithm $M$. The objective of every arm is to maximize the expected number of times it is pulled by the algorithm given by

$$u_i(M, \boldsymbol{\sigma}) := \mathbb{E}_M\big[n_T(i) \mid \boldsymbol{\sigma}\big],$$

where we condition on the arm strategies $\boldsymbol{\sigma}$ as these will (typically) impact the algorithm's decisions. We assume that the arms respond to the learning algorithm $M$ in Nash Equilibrium (NE).

**Definition 3.1** (Nash Equilibrium)**.** We say that $\boldsymbol{\sigma} = (\sigma_1, \dots, \sigma_K)$ forms a NE under the learner's algorithm $M$ if for all $i \in [K]$ and any deviating strategy $\sigma_i'$:

$$\mathbb{E}_M\big[n_T(i) \mid \sigma_i, \sigma_{-i}\big] \geq \mathbb{E}_M\big[n_T(i) \mid \sigma_i', \sigma_{-i}\big].$$

Let $\mathrm{NE}(M) := \{\boldsymbol{\sigma} : \boldsymbol{\sigma} \text{ is a NE under } M\}$ be the set of NE under algorithm $M$. We also consider $\varepsilon$-NE, which relax the requirement that no arm has an incentive to deviate.

**Definition 3.2** ($\varepsilon$-Nash Equilibrium)**.** We say that $\boldsymbol{\sigma} = (\sigma_1, \dots, \sigma_K)$ forms a $\varepsilon$-NE under algorithm $M$ if for all $i \in [K]$ and any deviating strategy $\sigma'$:

$$\mathbb{E}_M\big[n_T(i) \mid \sigma_i, \sigma_{-i}\big] \geq \mathbb{E}_M\big[n_T(i) \mid \sigma_i', \sigma_{-i}\big] - \varepsilon.$$

### 3.2 Strategic Regret

In the strategic linear contextual bandit, the performance of an algorithm depends on the arm strategies that it incentivizes. Naturally, minimizing regret when the arms always report their context truthfully is easier than when contexts are manipulated adversarially. We are interested in the *strategic regret* of an algorithm $M$ when the arms act according to a Nash equilibrium under $M$. Formally, for $\boldsymbol{\sigma} \in \mathrm{NE}(M)$ the strategic regret of $M$ is defined as

$$R_T(M, \boldsymbol{\sigma}) = \mathbb{E}_{M, \boldsymbol{\sigma}}\left[\sum_{t=1}^T \langle \theta^*, x_{t,i_t^*}^* \rangle - \langle \theta^*, x_{t,i_t}^* \rangle\right],$$

where $i_t^* = \mathrm{argmax}_{i \in [K]} \langle \theta^*, x_{t,i}^* \rangle$ is the optimal arm in round $t$. The regret guarantees of our algorithms hold uniformly over all NE that they induce, i.e., for $\max_{\boldsymbol{\sigma} \in \mathrm{NE}(M)} R_T(M, \boldsymbol{\sigma})$.

**Regularity Assumptions.** We allow for the true context vectors $x_{t,i}^*$ to be chosen adversarially by nature, and make the following assumptions about the linear contextual bandit model. We assume that both the context vectors and the rewards are bounded, i.e., $\max_{i,j \in [K]} \langle \theta^*, x_{t,i}^* - x_{t,j}^* \rangle \leq 1$ and $\|x_{t,i}^*\|_2 \leq 1$ for all $t \in [T]$. Moreover, we assume a constant optimality gap. That is, letting $\Delta_{t,i} := \langle \theta^*, x_{t,i_t^*}^* - x_{t,i}^* \rangle$, we assume that $\Delta := \min_{t,i:\Delta_{t,i}>0} \Delta_{t,i}$ is constant.

### 3.3 The Necessity of Mechanism Design

The first question that arises in this strategic setup is whether mechanism design, i.e., actively aligning the arms' incentives, is necessary to minimize regret. As expected, we find that this is the case. Standard algorithms, which are oblivious to the incentives they create, implicitly incentivize the arms to heavily misreport their contexts which makes minimizing regret virtually impossible.

We call a problem instance *trivial* if the algorithm that selects an arm uniformly at random every round achieves sublinear regret. Conversely, we call a problem instance *non-trivial* if the uniform selection suffers linear expected regret. We show that being incentive-unaware generally leads to linear regret in non-trivial instances (even when the learner has prior knowledge of $\theta^*$).

**Proposition 3.3.** *On any non-trivial problem instance, the incentive-unaware greedy algorithm that in round $t$ plays $i_t = \mathrm{argmax}_{i \in [K]} \langle \theta^*, x_{t,i} \rangle$ (with ties broken uniformly) suffers linear regret $\Omega(T)$ when the arms act according to any Nash equilibrium under the incentive-unaware greedy algorithm. Note that the incentive-unaware greedy algorithm has knowledge of $\theta^*$.*

**Mechanism 1:** The Greedy Grim Trigger Mechanism (GGTM)

---

1 **initialize:** $A_1 = [K]$
2 **for** $t = 1, \ldots, T$ **do**
3     Observe reported contexts $x_{t,1}, \ldots, x_{t,K}$
4     Play the (active) arm with largest reported reward: $i_t = \operatorname{argmax}_{i \in A_t} \langle \theta^*, x_{t,i} \rangle$
5     Observe reward $r_{t,i_t}$ from playing arm $i_t$.
6     **if** $\sum_{\ell \leq t : \, i_\ell = i_t} \langle \theta^*, x_{\ell, i_t} \rangle > \mathrm{UCB}_t(\hat{r}_{t,i_t})$ **then**
7         Eliminate arm $i_t$ from the active set: $A_{t+1} \leftarrow A_t \setminus \{i_t\}$.
8     **if** $A_{t+1} = \emptyset$ **then**
9         Stop playing any arm and receive zero reward for all remaining rounds.

---

*Similarly, algorithms for stochastic linear contextual bandits (LinUCB [1, 6]) and algorithms for linear contextual bandits with adversarial context corruptions (RobustBandit [8]) suffer linear regret when the arms act according to any Nash equilibrium that the algorithms incentivize.*

*Proof Sketch.* We demonstrate that the only NE for the arms lies in strategies that myopically maximize the probability of being selected in every round, which results in linear regret for the learner, because all arms always appear similarly good. The proof can be found in Appendix B. □

Another natural question to ask is whether exact incentive-compatibility is possible in the strategic linear contextual bandit. A learning algorithm is called *incentive-compatible* if truthfulness is a NE, i.e., reporting the true context $x_{t,i} = x^*_{t,i}$ every round is maximizing each arm's utility [14, 30]. For the interested reader, in Appendix A, we provide an incentive-compatible algorithm with constant regret in the *fully deterministic case*, where $\theta^*$ is known a priori as well as the rewards of pulled arms directly observable. However, when $\theta^*$ is unknown and/or the reward observations are subject to noise, we conjecture that exact incentive-compatibility (i.e., truthfulness is an exact NE, not $\varepsilon$-NE) is irreconcilable with regret minimization (cf. Appendix A).

## 4   Warm-Up: $\theta^*$ is Known in Advance

There are a number of challenges in the strategic linear contextual bandit. The most significant one is the need to incentivize the arms to be (approximately) truthful while simultaneously minimizing regret by learning about $\theta^*$ and selecting the best arms, even when observing (potentially) manipulated contexts. Notably, Proposition 3.3 showed that if we fail to align the arms' incentives, minimizing regret becomes impossible. Therefore, in the strategic linear contextual bandit, we must combine mechanism design with online learning techniques.

The uncertainty about $\theta^*$ poses a serious difficulty when trying to design such incentive-aware learning algorithms. As we only observe $x_{t,i}$ and $r_{t,i} = \langle \theta^*, x^*_{t,i} \rangle + \eta_t$, but do not observe the true context $x^*_{t,i}$, accurate estimation of $\theta^*$ is extremely challenging (and arguably intractable). We go into more depth in Section 5 when we introduce the Optimistic Grim Trigger Mechanism. For now, we consider the special case when $\theta^*$ is known to the learner in advance. This lets us highlight some of the challenges when connecting mechanism design with online learning in a less complex setting and introduce high-level ideas and concepts. When $\theta^*$ is known in advance, it can be instructive to consider what we refer to as the *reported (expected) reward* $\langle \theta^*, x_{t,i} \rangle$ instead of the *reported context vector* $x_{t,i}$ itself. Taking this perspective, when arm $i$ reports a $d$-dimensional vector $x_{t,i}$, we simply think of arm $i$ reporting a scalar reward $\langle \theta^*, x_{t,i} \rangle$. In what follows, it will prove useful to keep this abstraction in mind.[2]

### 4.1   The Greedy Grim Trigger Mechanism

One idea for a mechanism is to use a grim trigger. In repeated social dilemmas, the grim trigger strategy ensures cooperation among self-interested players by threatening with defection for all remaining rounds if the grim trigger condition is satisfied [28]. Typically, the grim trigger condition is defined so that it is immediately satisfied if a player defected at least once.

---

[2]We use the expressions 'reported reward' and 'expected reward' interchangeably to mean the reward we would expect to observe based on the context reported by the arm.

In the strategic contextual bandit, from the perspective of the learner, an arm can be considered to 'cooperate' if it is reporting its context truthfully. In turn, an arm 'defects' when it is reporting a gamed context. However, when an arm is reporting some context $x_{t,i}$ we do not know whether this arm truthfully reported its context or not, because we do not have access to the true context $x_{t,i}^*$. For this reason, we instead compare the expected reward $\langle \theta^*, x_{t,i} \rangle$ and the true reward $\langle \theta^*, x_{t,i}^* \rangle$. While we also cannot observe $\langle \theta^*, x_{t,i}^* \rangle$ directly, we do observe $r_{t,i} := \langle \theta^*, x_{t,i}^* \rangle + \eta_t$.

**Grim Trigger Condition.** Intuitively, if for any arm $i$ the total expected reward $\sum_{\ell \leq t:\ i_\ell = i} \langle \theta^*, x_{\ell,i} \rangle$ is larger than the total observed reward $\hat{r}_{t,i} := \sum_{\ell \leq t:\ i_\ell = i} r_{\ell,i}$, then arm $i$ must have been misreporting its contexts. However, $r_{\ell,i} := \langle \theta^*, x_{\ell,i}^* \rangle + \eta_\ell$ is random so that we instead use the *optimistic estimate* of the *observed reward* given by

$$\mathrm{UCB}_t(\hat{r}_{t,i}) := \sum_{\ell \leq t:\ i_\ell = i} r_{\ell,i} + 2\sqrt{n_t(i)\log(T)} \tag{1}$$

where $2\sqrt{n_t(i)\log(T)}$ is the confidence width which can be derived from Hoeffding's inequality. To implement the grim trigger, we then eliminate arm $i$ in round $t$ if the *total expected reward* is larger than the *optimistic estimate of the total observed reward*, i.e.,

$$\sum_{\ell \leq t:\ i_\ell = i} \langle \theta^*, x_{\ell,i} \rangle > \mathrm{UCB}_t(\hat{r}_{t,i}).$$

Note that using the optimistic estimate of the total observed reward ensures that elimination is justified with high probability. Conversely, we can guarantee with high probability that we do not erroneously eliminate a truthful arm.

**Selection Rule.** To complete the Greedy Grim Trigger Mechanism (GGTM, Mechanism 1), we then combine this with a greedy selection rule that pulls the arm with largest *reported reward* $\langle \theta^*, x_{t,i} \rangle$ in round $t$ from the set of arms that we believe have been truthful so far. Interestingly, even though we here assumed $\theta^*$ to be known in advance, we see that GGTM still utilizes online learning techniques such as the optimistic estimate (1) to align the arms' incentives.

It is also worth noting that—similar to its use in repeated social dilemmas—our grim trigger mechanism is *mutually destructive* in the sense that eliminating an arm for all remaining rounds is inherently bad for the learner (and of course for the eliminated arm as well).[3] Here lies the main challenge of the mechanism design and we must ensure that the arms are incentivized to "cooperate" (i.e., remain active) for a sufficiently long time.

### 4.2 Regret Analysis of GGTM

In what follows, we assume that each arm's strategy is restricted to reporting their 'reward' $\langle \theta^*, x_{t,i} \rangle$ not strictly lower than their true (mean) reward $\langle \theta^*, x_{t,i}^* \rangle$. It seems intuitive that no rational arm would ever under-report its value to the learner and make itself seem worse than it actually is. However, there are special cases, where under-reporting allows an arm to arbitrarily manipulate without detection. We discuss this later in Remark 4.3 and, more extensively, in Appendix C.

**Assumption 1.** We assume that $\langle \theta^*, x_{t,i} \rangle \geq \langle \theta^*, x_{t,i}^* \rangle$ for all $(t,i) \in [T] \times [K]$.

We now demonstrate that GGTM approximately incentivizes the arms to be truthful in the sense that the truthful strategy profile $\boldsymbol{\sigma}^*$ such that $x_{t,i} = x_{t,i}^*$ for all $(t,i) \in [T] \times [K]$ is an $\tilde{\mathcal{O}}(\sqrt{T})$-NE under GGTM. When the arms always report truthfully and no arm is erroneously eliminated, the greedy selection rule naturally selects the best arm every round so that GGTM's regret is constant.

**Theorem 4.1.** *Under the Greedy Grim Trigger Mechanism, being truthful is a $\tilde{\mathcal{O}}(\sqrt{T})$-NE for the arms. The strategic regret of GGTM when the arms act according to this equilibrium is at most*

$$R_T(\mathrm{GGTM}, \boldsymbol{\sigma}^*) \leq {}^1\!/_T.$$

*Proof Sketch.* By design of the grim trigger, it is straightforward to show that the probability that a truthful arm gets eliminated is at most ${}^1\!/_{T^2}$. Moreover, the grim trigger ensures that no arm can 'poach' selections from a truthful arm more than order $\sqrt{T}$ times by misreporting its contexts. This achieves two things: (a) it protects truthful arms and guarantees that truthfulness is a good strategy, and (b) limits an arm's profit from being untruthful. The proof can be found in Appendix D.2. □

---

[3] Note that in linear contextual bandits there is no single optimal arm, but the optimal arm changes per round.

Theorem 4.1 tells us that truthfulness is an approximate NE. We now also provide a more holistic strategic regret guarantee of $\tilde{\mathcal{O}}(K^2\sqrt{KT})$ in *every* Nash equilibrium under GGTM. Proving this is more complicated as the arms can profit from exploiting our uncertainty about their truthfulness (i.e., the looseness of the grim trigger).

**Theorem 4.2.** *The Greedy Grim Trigger Mechanism has strategic regret*

$$R_T(\text{GGTM}, \boldsymbol{\sigma}) = \mathcal{O}\left(\underbrace{\sqrt{KT\log(T)}}_{\text{cost of manipulation}} + \underbrace{K^2\sqrt{KT\log(T)}}_{\text{cost of mechanism design}}\right) \tag{2}$$

*for every* $\boldsymbol{\sigma} \in \text{NE(GGTM)}$. *Hence,* $\max_{\boldsymbol{\sigma}\in\text{NE(GGTM)}} R_T(\text{GGTM}, \boldsymbol{\sigma}) = \tilde{\mathcal{O}}\left(K^2\sqrt{KT}\right)$.

*Proof Sketch.* The regret analysis is notably more complicated than the one in Theorem 4.1, as we must bound the regret due to the arms exploiting our uncertainty as well as the cost of committing to the grim trigger. Both of these quantities do not play a role when the arms always report truthfully (like in Theorem 4.1). A complete proof can be found in Appendix D. $\qquad\square$

The regret bound (2) suggests that there are two sources of regret. The first term is due to our mechanism design being approximate (relying on estimates), which leaves room for the arms to exploit our uncertainty and misreport their contexts to obtain additional selections. The second part of (2) is the cost of the mechanism design, i.e., the cost of committing to the grim trigger. We suffer constant regret any round in which the round-optimal arm is no longer in the active set. In the worst-case, this quantity is of order $K^2\sqrt{KT}$.

**Remark 4.3.** *We want to briefly comment on Assumption 1. It appears intuitive that any rational arm would never under-report its value, i.e., make itself look worse than it actually is. However, in Appendix C, we provide a simple example where occasionally under-reporting its value allows an arm to simulate an environment where it is always optimal, even though it is in fact only optimal half of the time. We will explain in the example that without additional strong assumptions on the noise distribution the two environments are indistinguishable so that such manipulation by the arms appears unavoidable when trying to maximize rewards.*

## 5 The Optimistic Grim Trigger Mechanism

The problem of estimating the unknown parameter $\theta^*$ appears daunting given that the arms can strategically alter their contexts to manipulate our estimate of $\theta^*$ to their advantage. In fact, imagine an arm manipulating its contexts orthogonal to $\theta^*$ so that $\langle\theta^*, x_{t,i} - x^*_{t,i}\rangle = 0$ but $x_{t,i} \neq x^*_{t,i}$. Observing only $x_{t,i}$ and $r_{t,i} := \langle\theta^*, x^*_{t,i}\rangle + \eta_t$, our estimate of $\theta^*$ becomes biased and could be arbitrarily far off the true parameter $\theta^*$ even though the gamed context and true context have the same reward w.r.t. $\theta^*$. This is also the case more generally. Since we observe neither $\theta^*$ nor $x^*_{t,i}$, any observed combination of $x_{t,i}$ and $r_{t,i}$ will "make sense" to us. *But, how can we incentivize the arms to report truthfully and minimize regret despite incorrect estimates of $\theta^*$?*

Our key observation is that learning $\theta^*$ is not necessary to incentivize the arms or minimize regret; it appears to be a hopeless endeavour after all. The idea of the Optimistic Grim Trigger Mechanism (OptGTM, Mechanism 2) is to construct pessimistic estimates of the total reward we expected from pulling an arm. Importantly, we can construct such pessimistic estimates of the expected (i.e., "reported") reward even when the contexts are manipulated. OptGTM then threatens arms with elimination if our *pessimistic* estimate of the expected reward exceeds the *optimistic* estimate of the observed reward. Interestingly, this does not relate to the amount of corruption in the feature space and, in fact, $\sum_{t,i}\|x_{t,i} - x^*_{t,i}\|_2$ could become arbitrarily large. However, it does bound the effect of each arm's strategic manipulation on the decisions we make and thereby allows for effective incentive design and regret minimization.

To construct pessimistic (and optimistic) estimates of the expected reward, we use independent estimators $\hat{\theta}_{t,i}$ and confidence sets $C_{t,i}$ around $\hat{\theta}_{t,i}$, which do not take into account that the contexts are potentially manipulated. That is, we have a separate estimator and confidence set for each arm $i \in [K]$. This will prevent one arm influencing the elimination of another. It also stops collusive arm behavior, where a majority group of the arms could dominate and steer our estimation process.

**Mechanism 2:** The Optimistic Grim Trigger Mechanism (OptGTM)

---

**1**   **initialize:** $A_1 = [K]$
**2**   **for** $t = 1, \ldots, T$ **do**
**3**      Observe reported contexts $\mathcal{X}_t = \{x_{t,1}, \ldots, x_{t,K}\}$.
**4**      Play the active arm with largest reported optimistic reward

$$i_t = \operatorname*{argmax}_{i \in A_t} \mathrm{UCB}_{t,i}(x_{t,i}).$$

**5**      Receive reward $r_{t,i_t}$ from playing arm $i_t$.
**6**      **if** $\sum_{\ell \leq t:\, i_\ell = i_t} \mathrm{LCB}_{\ell,i_t}(x_{\ell,i_t}) > \mathrm{UCB}_t(\hat{r}_{t,i_t})$ **then**
**7**         Eliminate arm $i_t$ from the active set: $A_{t+1} \leftarrow A_t \setminus \{i_t\}$.
**8**      **if** $A_{t+1} = \emptyset$ **then**
**9**         Stop playing any arm and receive zero reward for all remaining rounds.

---

**Confidence Sets.**   For every arm $i \in [K]$ we define the least-squares estimator $\hat{\theta}_{t,i}$ w.r.t. its *reported* contexts and observed rewards before round $t$ as

$$\hat{\theta}_{t,i} = \operatorname*{argmin}_{\theta \in \mathbb{R}^d} \left( \sum\nolimits_{\ell < t:\, i_\ell = i} \left( \langle \theta, x_{\ell,i} \rangle - r_{\ell,i} \right)^2 + \lambda \|\theta\|_2^2 \right), \tag{3}$$

where $\lambda > 0$. We then define the confidence set $C_{t,i} := \{\theta \in \mathbb{R}^d : \|\hat{\theta}_{t,i} - \theta\|_{V_{t,i}}^2 \leq \beta_{t,i}\}$ where $\beta_{t,i} := \mathcal{O}(d \log(n_t(i)))$ is the confidence size. Here, $V_{t,i}$ is the covariance matrix of reported contexts of arm $i$ given by $V_{1,i} := \lambda I$ and $V_{t,i} := \lambda I + \sum_{\ell < t:\, i_\ell = i} x_{\ell,i} x_{\ell,i}^\top$.[4]

It is well-known that if the contexts were always reported truthfully, i.e., $x_{t,i} = x_{t,i}^*$, then with high probability $\theta^* \in C_{t,i}$. But, if the sequence of reported contexts $x_{t,i}$ substantially differs from the true contexts $x_{t,i}^*$ there is no (high probability) guarantee that the true parameter $\theta^*$ is element in $C_{t,i}$. In the literature on learning with adversarial corruptions (in linear contextual bandits), the standard approach to deal with this is to widen the confidence set. However, for this to be effective we would need to assume a sufficiently small corruption budget for the arms and prior knowledge of the total amount of corruption, both of which we explicitly do not assume here.

Slightly overloading notation, we instead define the optimistic and pessimistic estimate of the expected reward of a context vector $x$ w.r.t. arm $i$ as

$$\mathrm{UCB}_{t,i}(x) := \langle \hat{\theta}_{t,i}, x \rangle + \sqrt{\beta_{t,i}} \|x\|_{V_{t,i}^{-1}} \quad \text{and} \quad \mathrm{LCB}_{t,i}(x) := \langle \hat{\theta}_{t,i}, x \rangle - \sqrt{\beta_{t,i}} \|x\|_{V_{t,i}^{-1}}.$$

We chose to state these estimates using additive bonuses. However, it can be convenient to think of them as $\mathrm{UCB}_{t,i}(x) \approx \max_{\theta \in C_{t,i}} \langle \theta, x \rangle$ and $\mathrm{LCB}_{t,i}(x) \approx \min_{\theta \in C_{t,i}} \langle \theta, x \rangle$.

**Grim Trigger Condition.**   In round $t \in [T]$, we eliminate arm $i$ from $A_t$ if the pessimistic estimate using the reports is larger than the optimistic estimate using the total observed reward, i.e.,

$$\sum_{\ell \leq t:\, i_\ell = i} \left( \langle \hat{\theta}_{\ell,i}, x_{\ell,i} \rangle - \sqrt{\beta_\ell} \|x_{\ell,i}\|_{V_{\ell,i}^{-1}} \right) > \sum_{\ell \leq t:\, i_\ell = i} r_{\ell,i} + 2\sqrt{n_t(i) \log(T)}. \tag{4}$$

In other words, $\sum_{\ell \leq t:\, i_\ell = i} \mathrm{LCB}_{\ell,i}(x_{\ell,i}) > \mathrm{UCB}_t(\hat{r}_{t,i})$.

Examining the left side of (4), the careful reader may wonder why we do not simply use our latest and arguably best estimate $\hat{\theta}_{t,i}$, but instead the whole sequence of "out-dated" estimators $\hat{\theta}_{\ell,i}$ from previous rounds. In fact, this is crucial for the grim trigger. Using $\hat{\theta}_{t,i}$ renders the grim trigger condition ineffective, because, by definition, $\hat{\theta}_{t,i}$ is the (least-squares) minimizer (3) of the difference between $\sum_{\ell \leq t:\, i_\ell = i} \langle \hat{\theta}_{t,i}, x_{\ell,i} \rangle$ and $\sum_{\ell \leq t:\, i_\ell = i} r_{\ell,i}$. Hence, when using $\hat{\theta}_{t,i}$ the grim trigger condition may not be satisfied even when the arms significantly and repeatedly misreport their contexts.

**Selection Rule.**   We complete the OptGTM algorithm by selecting arms optimistically with respect to each arm's own estimator and confidence set. That is, OptGTM selects the active arm with maximal optimistic (expected) reward $\mathrm{UCB}_{t,i}(x_{t,i}) := \langle \hat{\theta}_{t,i}, x_{t,i} \rangle + \sqrt{\beta_{t,i}} \|x_{t,i}\|_{V_{t,i}^{-1}}$ in round $t$. We see that

---

[4]For more details on the design of least-squares estimators and the confidence sets, we refer to Abbasi-Yadkori et al. [1] and Lattimore and Szepesvári [24] (Chapter 20).

the grim trigger (4) incentivizes arms to ensure that over the course of all rounds $\text{LCB}_{t,i}(x_{t,i})$ is not much smaller than $r_{t,i} := \langle \theta^*, x_{t,i}^* \rangle + \eta_t$. Hence, $\text{UCB}_{t,i}(x_{t,i})$ is also not substantially smaller than the true mean reward $\langle \theta^*, x_{t,i}^* \rangle$. This suggests that playing optimistically is a good strategy for the learner as long as the selected arm does not satisfy (4).

## 5.1 Regret Analysis of OptGTM

When $\theta^*$ was known to the learner in advance, we assumed that the arms never report a value smaller than their true value, i.e., $\langle \theta^*, x_{t,i} \rangle \geq \langle \theta^*, x_{t,i}^* \rangle$ for all $(t, i) \in [T] \times [K]$. Now, when $\theta^*$ is unknown to the learner, we similarly assume that the arms do not report their optimistic value less than their true value. Again, it seems intuitive that in any given round, no arm would under-report its worth.

**Assumption 2.** We assume that $\max_{\theta \in C_{t,i}} \langle \theta, x_{t,i} \rangle \geq \langle \theta^*, x_{t,i}^* \rangle$ for all $(t, i) \in [T] \times [K]$.

We find that OptGTM approximately incentivizes the arms to be truthful and, when the arms report truthfully, OptGTM suffers at most $\tilde{\mathcal{O}}(d\sqrt{KT})$ regret.

**Theorem 5.1.** *Under the Optimistic Grim Trigger Mechanism, being truthful is a $\tilde{\mathcal{O}}(d\sqrt{KT})$-NE. When the arms report truthfully, the strategic regret of OptGTM under this approximate NE is at most*

$$R_T(\text{OptGTM}, \boldsymbol{\sigma}^*) = \tilde{\mathcal{O}}(d\sqrt{KT}).$$

The optimal regret in standard non-strategic linear contextual bandits is $\Theta(d\sqrt{T})$ so that OptGTM is optimal up to a factor of $\sqrt{K}$ (and logarithmic factors) when the arms report truthfully. The additional factor of $\sqrt{K}$ is caused by the fact that OptGTM maintains independent estimates for each arm. We now also provide a strategic regret bound for every NE of the arms under OptGTM.

**Theorem 5.2.** *The Optimistic Grim Trigger Mechanism has strategic regret*

$$R_T(\text{OptGTM}, \boldsymbol{\sigma}) = \mathcal{O}\left(d\log(T)\sqrt{KT} + d\log(T)K^2\sqrt{KT}\right).$$

*for every $\boldsymbol{\sigma} \in \text{NE}(\text{OptGTM})$. Hence, $\max_{\boldsymbol{\sigma} \in \text{NE}(\text{OptGTM})} R_T(\text{OptGTM}, \boldsymbol{\sigma}) = \tilde{\mathcal{O}}\left(dK^2\sqrt{KT}\right).$*

The proof ideas of Theorem 5.1 and Theorem 5.2 are similar to their counterparts in Section 4. The main difference lies in a more technically challenging analysis of the grim trigger condition (4). We also see that in contrast to non-strategic linear contextual bandits, where the regret typically does not depend on the number of arms $K$, Theorem 5.1 and Theorem 5.2 include a factor of $\sqrt{K}$ and $K^2\sqrt{K}$, respectively. A dependence on $K$ is expected due to the strategic nature of the arms which forces us to explicitly incentivize each arm to be truthful. However, we conjecture that the regret bound in Theorem 5.2 is not tight in $K$ and expect the optimal dependence on the number of arms to be $\sqrt{K}$. The proofs of Theorem 5.1 and Theorem 5.2 can be found in Appendix E.

## 6 Experiments: Simulating Strategic Context Manipulation

We here experimentally analyze the efficacy of OptGTM when the arms strategically manipulate their contexts in response to our learning algorithm. We compare the performance of OptGTM with the traditional LinUCB algorithm [1, 6], which—as shown in Proposition 3.3—implicitly incentivizes the arms to manipulate their contexts and suffers large regret when the arms are strategic.

Contrary to the assumption of arms playing NE, we here model strategic arm behavior by letting the arms update their strategy (i.e., what contexts to report) based on past interactions with the algorithms. More precisely, we assume that the strategic arms interact with the deployed algorithm (i.e., OptGTM or LinUCB) over the course of 20 epochs, with each epoch consisting of $T = 10k$ rounds. At the end of each epoch, every arm then updates its strategy using gradient ascent w.r.t. its utility. Importantly, this approach requires no prior knowledge from the arms, as they learn entirely through sequential interaction. This does not necessarily lead to equilibrium strategies, but serves as a natural model of strategic gaming behavior under which to study the algorithms.

**Experimental Setup.** We associate each arm with a true feature vector $y_i^* \in \mathbb{R}^{d_1}$ (e.g., product features) and randomly sample a sequence of user vectors $c_t \in \mathbb{R}^{d_2}$ (i.e., customer features). We assume that every arm can alter its feature vector $y_i^*$ by reporting some other vector $y_i$, but cannot alter the user contexts $c_t$. We use a feature mapping $\varphi(c_t, y_i) = x_{t,i}$ to map $y_i \in \mathbb{R}^{d_1}$ and $c_t \in \mathbb{R}^{d_2}$ to an arm-specific context $x_{t,i} \in \mathbb{R}^d$ that the algorithm observes. At the end of every epoch, each arm then performs an approximated gradient step on $y_i$ w.r.t. its utility, i.e., the number of times it is selected. We let $K = 5$ and $d = d_1 = d_2 = 5$ and average the results over 10 runs.

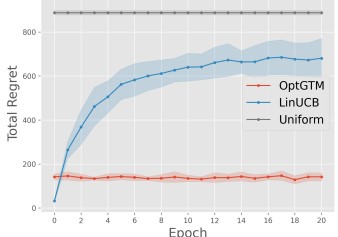
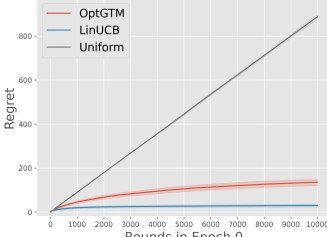
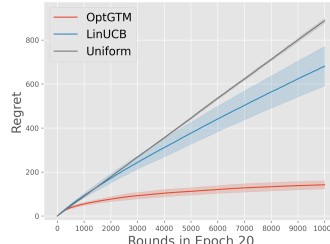

(a) Total strategic regret $R_T$ as the arms adapt their strategies to the deployed algorithm over the course of 20 epochs.

(b) Epoch 0 (Truthful Arms): Regret as a function of $t$ *before* the arms have interacted with the deployed algorithm.

(c) Epoch 20 (Strategic Arms): Regret as a function of $t$ *after* the arms have interacted with the deployed algorithm.

Figure 1: Comparison of the strategic regret of OptGTM and LinUCB. The strategic arms adapt their strategies gradually over the course of 20 epochs. OptGTM performs similarly across all epochs, whereas LinUCB performs increasingly worse as the arms adapt to the algorithm (Figure 1a). Figure 1b and 1c provide a closer look at the regret of the algorithms across the $T$ rounds in the initial epoch, where the arms are truthful, and the final epoch after the arms have adapted to the algorithms.

**Results.** In Figure 1a, we observe that OptGTM performs similarly well across all epochs, which suggests that OptGTM successfully discourages the emergence of harmful gaming behavior. In contrast, as the arms adapt their strategies (i.e., what features to report), LinUCB suffers increasingly more regret and almost performs as badly as uniform sampling in the final epoch. In epoch 0, when the all arms are

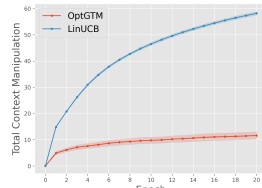
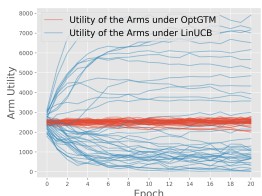

Figure 2: Context manipulation $\sum_{t,i} \|x_{t,i}^* - x_{t,i}\|_2$.

Figure 3: Utility of the arms for each of the 10 runs.

truthful, i.e., are non-strategic, LinUCB performs better than OptGTM (Figure 1b). This is expected as OptGTM suffers additional regret due to maintaining independent estimates of $\theta^*$ for each arm (as a mechanism to incentivize truthfulness). However, OptGTM significantly outperforms LinUCB as the arms strategically adapt, which is most evident in the final epoch (Figure 1c). Interestingly, as already suggested in Section 5, OptGTM cannot prevent manipulation in the feature space (see Figure 2). However, OptGTM does manage to bound the effect of the manipulation on the regret (Figure 1a) and, most importantly, the effect on the utility of the arms as well (Figure 3). As a result, the arms are discouraged from heavily gaming their contexts and the context manipulation has only a minor effect on the actions taken by OptGTM.

## 7 Discussion

We study a strategic variant of the linear contextual bandit problem, where the arms strategically misreport privately observed contexts to maximize their selection frequency. To address this, we design two online learning mechanisms: the Greedy and the Optimistic Grim Trigger Mechanism, for the scenario where $\theta^*$ is known and unknown, respectively. We demonstrate that our mechanisms incentivize the arms to be approximately truthful and, in doing so, effectively minimize regret. More generally, with this work, we aim to advance our understanding of problems at the intersection of online learning and mechanism design. As the digital landscape, including online platforms and marketplaces, becomes increasingly agentic and dominated by self-interested agents, it will be crucial to understand the incentives created by learning algorithms and to align these incentives while optimizing for performance.

**Limitations.** One limitation is the otherwise intuitive assumption that the arms do not under-report their value to the learner (Assumption 1 and Assumption 2). Secondly, we believe that the factor of $K^2$ in the universal regret guarantees of Theorem 4.2 and Theorem 5.2 is suboptimal and we conjecture that the optimal worst-case strategic regret is given by $\mathcal{O}(d\sqrt{KT})$. We leave this investigation for future work.

## Acknowledgements

Thomas Kleine Buening is supported by the UKRI Prosperity Partnership Scheme (Project FAIR). Haifeng Xu is supported in part by the Army Research Office Award W911NF-23-1-0030, the ONR Award N00014-23-1-2802 and the NSF Award CCF-2303372.

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

# A Remarks on Incentive-Compatible No-Regret Algorithms

In Section 3, we conjectured that there exists no incentive-compatible no-regret algorithm when the reward observations are subject to noise and $\theta^*$ unknown. For the interested reader, we here consider the *fully deterministic case* where $\theta^*$ is known a priori and reward observations are directly observable, i.e., subject to no noise so that $\eta_t \equiv 0$. We can design the following provably incentive-compatible no-regret algorithm. In fact, we show that this mechanism is strategyproof, i.e., incentive-compatible in weakly dominant strategies.

---

**Mechanism 3:** Incentive-Compatible No-Regret Algorithm in the Fully Deterministic Case

---

1 **initialize:** $A_1 = [K]$
2 **for** $t < T - (K + 1)$ **do**
3      Play $i_t \in \operatorname{argmax}_{i \in A_t} \langle \theta^*, x_{t,i} \rangle$
4      Observe reward $r^*_{t,i_t} := \langle \theta^*, x^*_{t,i_t} \rangle$     (i.e., rewards of chosen arms are directly observable)
5      **if** $\langle \theta^*, x_{t,i_t} \rangle \neq r^*_{t,i_t}$ **then**
6          Eliminate arm $i_t$: $A_{t+1} \leftarrow A_t \setminus \{i_t\}$.
7 **for** $t \geq T - (K + 1)$ **do**
8      Play $i_t \sim \mathrm{Uniform}(A_t)$
9      Observe reward $r^*_{t,i_t} := \langle \theta^*, x^*_{t,i_t} \rangle$
10      **if** $\langle \theta^*, x_{t,i_t} \rangle \neq r^*_{t,i_t}$ **then**
11          Eliminate arm $i_t$: $A_{t+1} \leftarrow A_t \setminus \{i_t\}$.

---

**Lemma A.1.** *Mechanism 3 is strategyproof, i.e., being truthful is a weakly dominant strategy for every arm. Moreover, Mechanism 3 suffers at most $K + 1$ strategic regret in every Nash equilibrium of the arms.*[5]

*Proof.* **Incentive-Compatibility in Weakly Dominant Strategies.** It is easy to see that for the last $K + 1$ rounds, reporting truthfully, i.e., reporting $x^*_{t,i}$, is a weakly dominant strategy, since the the set of active arms is played uniformly and nothing can be gained from misreporting (an arm can only miss out on being selected by misreporting in the last $K + 1$ steps). Hence, conditional on any history, reporting truthfully is the best continuation for any arm. In particular, when an arm plays truthfully in these rounds the obtained utility in the last $K + 1$ steps is at least $\frac{K+1}{K}$, since $|A_t| \leq K$.

Now, for the time steps $t < T - (K + 1)$ note that any untruthful strategy can obtain at most one more allocation than the truthful strategy, because if $i_t = i$ and $\langle \theta^*, x_{t,i} \rangle > \langle \theta^*, x^*_{t,i} \rangle$, then arm $i$ is eliminated immediately. Hence, at most utility 1 can be gained from receiving an allocation by misreporting. However, in this case the arm gets eliminated and receives utility 0 in the last $K + 1$ rounds. As seen before the minimum utility the truthful strategy receives in the last $K + 1$ receives is $\frac{K+1}{K} > 1$. Consequently, irrespective of the other arms strategies, the truthful strategy is (weakly) optimal for arm $i$.

One may wonder why the truthful strategy is not *strictly* dominant. To see this note that reporting any $x_{t,i} \neq x^*_{t,i}$ such that the difference $x_{t,i} - x^*_{t,i}$ is orthogonal to $\theta^*$, i.e., $\langle \theta^*, x_{t,i} - x^*_{t,i} \rangle = 0$, does not cause elimination and is equivalent under Mechanism 3. In other words, such untruthful strategies, which however have no effect on the selection, are equally good.

**Regret.** The regret in the last $K + 1$ rounds is trivially bounded by $K + 1$. When showing that the algorithm is strategyproof we showed that any untruthful strategy such that there exists $i_t = i$ with $\langle \theta^*, x_{t,i} \rangle > \langle \theta^*, x^*_{t,i} \rangle$ is worse than the truthful strategy independently from what the other arms are playing. Hence, in any Nash equilibrium arm $i$ chooses strategies such that if $i_t = i$ then $\langle \theta^*, x_{t,i} \rangle = \langle \theta^*, x^*_{t,i} \rangle$. In other words, since the selection is greedy, Mechanism 3 selected the best arm in the given round. Mechanism 3 therefore suffers zero regret in the first $T - (K + 1)$ regret in any Nash equilibrium of the arms. $\qquad\square$

As discussed in Section 3, we conjecture that there exists no incentive-compatible no-regret algorithm for the strategic linear contextual bandits when the reward observations are subject to noise. The

---

[5]Note that since truthfulness is only weakly dominant there could be other Nash equilibria.

intuition for this conjecture is as follows. Suppose there exists a learning algorithm $M$ that is incentive-compatible and no-regret, that is, the strategy profile where every arm is *always* truthful is a NE. Since $M$ is also no-regret, the selection policy of $M$ must depend on the reported contexts in some way. In particular, in some round $t$ in which $M$ does not select arm $i$—but $M$ maps from reported contexts to an action in $[K]$—there must exist a context $\tilde{x}_t$ that arm $i$ could report that increases its probability of being selected.

Suppose arm $i$ changes its strategy from $\sigma_i^*$ (i.e., being truthful) to the strategy that is always truthful except for round $t$ where it reports $\tilde{x}_t$ instead of $x_{t,i}^*$. The algorithm $M$ then observes a reward drawn from a distribution with mean $\langle \theta^*, x_{t,i}^* \rangle$, but might have expected a reward drawn from a distribution with mean $\langle \theta^*, \tilde{x}_t \rangle$. We believe that the difference in observed and expected reward is statistically insignificant when arm $i$ only misreports a single or constant number of times. However, due to the intricate relationship between the learning algorithm and the induced NE strategies for the $K$ arms, providing a rigorous argument for this is challenging.

# B  Proof of Proposition 3.3

*Proof of Proposition 3.3.* We begin with the incentive-unaware greedy algorithm that in round $t$ pulls arm $i_t = \operatorname{argmax}_{i \in [K]} \langle \theta^*, x_{t,i} \rangle$. Let $\tilde{x} := \operatorname{argmax}_{\|x\| \leq 1} \langle \theta^*, x \rangle$ and w.l.o.g. we assume that $\tilde{x}$ is unique. We show that the strategy profile, where every arm always reports $\tilde{x}$ is the only Nash equilibrium under the incentive-unaware greedy algorithm. Let $\boldsymbol{\sigma}$ be any strategy profile which is such that there exists a round $t$ and arm $i$ such that $x_{t,i} \neq \tilde{x}$. We distinguish between two cases.

**Case 1:**  There exists a round $t$ and arm $i$ such that $\langle \theta^*, x_{t,i} \rangle < \max_{j \in [K]} \langle \theta^*, x_{t,j} \rangle$.

Note that this implies that arm $i$ is not selected by the learner. However, by reporting $\tilde{x}$ instead of $x_{t,i}$, arm $i$ is guaranteed to be selected with probability at least $1/K$. Hence, reporting $x_{t,i}$ is strictly worse than reporting $\tilde{x}$ so that $\boldsymbol{\sigma}$ cannot be a NE.

**Case 2:**  $\langle \theta^*, x_{t,i} \rangle = \max_{j \in [K]} \langle \theta^*, x_{t,j} \rangle$ for all rounds $t$ and arms $i$.

Note that this implies that each arm $i$ is selected with probability $1/K$ every round.[6] Suppose that for any of these rounds $t$ we have $\max_{j \in [K]} \langle \theta^*, x_{t,j} \rangle < \langle \theta^*, \tilde{x} \rangle$. Then, by reporting $\tilde{x}$ instead of $x_{t,i}$ arm $i$ could ensure to be selected with probability one. Hence, the strategy where arm $i$ in round $t$ reports $\tilde{x}$ instead of $x_{t,i}$ is a strictly better response. Therefore, $\boldsymbol{\sigma}$ cannot be a NE. The other case is when $\max_{j \in [K]} \langle \theta^*, x_{t,j} \rangle = \langle \theta^*, \tilde{x} \rangle$, but this cannot be because $\boldsymbol{\sigma}$ is supposed to be different to always reporting $\tilde{x}$.

Consequently, the strategy profile where every arm always reports $\tilde{x}$ is the only NE under the incentive-unaware greedy algorithm. Under this strategy profile, the incentive-unaware greedy algorithm will play uniformly and therefore suffer linear regret.

**Insufficiency of Non-Strategic Linear Contextual Bandit Algorithms.**  It is not really surprising that algorithms for non-strategic linear contextual bandits fail in the strategic linear contextual bandit, since such algorithms implicitly incentivize the arms to "compete" in every round by misreporting their context as the best possible one. Nothing prevents the arms to not myopically optimize their probability of being selected every round. As an example of a standard algorithm for non-strategic linear contextual bandits we consider LinUCB that in the non-strategic problem setup enjoys a regret guarantee of $\tilde{\mathcal{O}}(d\sqrt{T})$.

The reasons for LinUCB's failure in this strategic problem are the same as for the incentive-unaware greedy algorithm from before. It will be the strictly dominant strategy for the arms to maximize their selection probability in the given round by misreporting their context. Recall that LinUCB maintains a least-squares estimator given by

$$\hat{\theta}_t = \operatorname*{argmin}_{\theta \in \mathbb{R}^d} \sum_{\ell=1}^{t-1} (\langle \theta, x_{\ell,i_\ell} \rangle - r_{\ell,i_\ell})^2 + \lambda \|\theta\|_2^2$$

---

[6]This already implies linear regret, but it will be instructive to still show that the only NE is in maximally gaming strategies.

and in round $t$ selects arm (ties broken uniformly at random)

$$i_t = \operatorname*{argmax}_{i \in [K]} \langle \hat{\theta}_t, x_{t,i} \rangle + \sqrt{\beta_t} \|x_{t,i}\|_{V_t^{-1}},$$

where $\beta_t \approx d \log(T)$ and $V_t = \lambda I + \sum_{i=1}^{t-1} x_{\ell,i_\ell} x_{\ell,i}^\top$. Let $\mathrm{UCB}_t(x) := \langle \hat{\theta}_t, x \rangle + \sqrt{\beta_t} \|x\|_{V_t^{-1}}$.

The argument for LinUCB will be the same as for the incentive-unaware greedy algorithm. Let $\tilde{x}_t := \operatorname{argmax}_{\|x\|_2 \leq 1} \mathrm{UCB}_t(x)$ and w.l.o.g. assume that $\tilde{x}_t$ is unique.

Importantly, in what follows, keep in mind that it will not matter how the reports of arm $i$ influenced $\hat{\theta}_t$ or $V_t$ in previous rounds. Once again, suppose $\boldsymbol{\sigma}$ is a strategy profile such that there exists a round $t$ and arm $i$ such that conditioned on the history $x_{t,i} \neq \tilde{x}_t$. Once again we distinguish between the following two cases:

**Case 1:** There exists a round $t$ and arm $i$ such that $\mathrm{UCB}_t(x_{t,i}) < \max_{j \in [K]} \mathrm{UCB}_t(x_{t,j})$.

This implies that arm $i$ was not selected by the learner in round $t$. However, by reporting $\tilde{x}_t$ and in all future rounds report $\tilde{x}_\ell$ for $\ell > t$, arm $i$ can guarantee to be selected with probability at least $1/K$ in round $t$ and at least as many selections as under $\sigma_i$. Hence, $\sigma_i$ cannot be a best response to $\sigma_{-i}$.

**Case 2:** $\mathrm{UCB}_t(x_{t,i}) = \max_{j \in [K]} \mathrm{UCB}_t(x_{t,j})$ for all rounds $t$ and arms $i$.

Note that this implies that arm $i$ is selected with probability $1/K$ every round. Suppose that for any round $t$ it is the case that $\max_{j \in [K]} \mathrm{UCB}_t(x_{t,j}) < \mathrm{UCB}_t(\tilde{x})$. Then, by choosing strategy $\tilde{x}_t$ in round $t$ and $\tilde{x}_\ell$ adaptively for all future rounds $\ell > t$, arm $i$ obtains more selections than when reporting $x_{t,i}$. Hence, $\boldsymbol{\sigma}$ cannot be a NE.

As a result, the strategy profile where all arms report $\tilde{x}_t$ in round $t$ is the only NE and LinUCB suffers linear regret, as it pulls arms uniformly at random. In exactly the same way, we can also show that the algorithms for linear contextual bandits with adversarial context corruptions in [8] suffer linear regret. $\qquad\square$

## C  Assumption 1 and Remark 4.3

**Example 1.** We here give a simple example where a strategic arm can simulate a situation where it is always optimal even though it is only optimal half of the time.

Let $\theta^* = 1$ and consider the following problem instance with two arms 1 and 2, where

$$x_{t,1}^* = \begin{cases} 0, & t \text{ is even} \\ 1, & t \text{ is odd} \end{cases} \quad \text{and} \quad x_{t,2}^* = 1/4.$$

Now, suppose that arm 1 always reports $x_{t,1} = 1/2$ and arm 2 reports truthfully (or approximately so). Then, arm 1 appears optimal every round $t$. In particular, on average arm when we pull arm 1 it has reward $1/2$, which is consistent with its report of $x_{t,1} = 1/2$.

Now, consider a second problem instance, where

$$x_{t,1}^* = 1/2 \quad \text{and} \quad x_{t,2}^* = 1/4.$$

Recall that we assume that, like almost always in the literature, the noise is sub-Gaussian. As an example, let's consider Bernoulli-noise such that $\mathbb{P}(r_{t,i} = 1) = x_{t,i}^* = 1 - \mathbb{P}(r_{t,i} = 0)$. Then, the first environment when arm 1 manipulates as suggested is identical to the second environment when the arms are truthful. In the second environment, to suffer sublinear regret we must select arm 1 order $T$ many times. However, in the first environment, we must select arm 1 only order $o(T)$ many times.

**Discussion.** Based on these observations, we expect that we would have to make additional (strong) assumptions about the distribution of the noise and the prior knowledge of the learner in order to drop Assumption 1. As an example, let's assume standard normal noise $\mathcal{N}(0, 1)$ and that the learner knows that the variance is always 1 a priori. Then, one potentially effective approach would be to extend our current grim trigger to additionally threaten arms that misreport their variance. Of course, the variance is unknown, however, we could estimate the variance of each arm's reports separately and use confidence intervals around the estimated variance. We could then threaten an arm with elimination if the arm's estimated variance falls out of the confidence interval. However, we expect there to be several technical subtleties in analyzing such variance-aware mechanisms.

# D  Proof of Theorem 4.1 and Theorem 4.2

## D.1  Preliminaries

We begin with some preliminaries that we use in the proofs of both Theorem 4.1 and Theorem 4.2. Note that $\mathbb{E}[r_{t,i}] = \langle \theta^*, x_{t,i}^* \rangle$ and we denote the total true mean reward by

$$\hat{r}_{t,i}^* := \sum_{\ell \leq t \, : \, i_\ell = i} \langle \theta^*, x_{\ell,i}^* \rangle.$$

Let us also recall the definition of the total observed reward as $\hat{r}_{t,i} := \sum_{\ell \leq t \, : \, i_\ell = i} r_{\ell,i}$ and recall its upper confidence bound $\mathrm{UCB}_t(\hat{r}_{t,i}) := \hat{r}_{t,i} + 2\sqrt{n_t(i)\log(T)}$ and define the lower confidence bound $\mathrm{LCB}_t(\hat{r}_{t,i}) := \hat{r}_{t,i} - 2\sqrt{n_t(i)\log(T)}$.

We now analyze the basic properties of the grim trigger condition of GGTM, which eliminates arm $i$ in round $t$ if

$$\sum_{\ell \leq t \, : \, i_\ell = i} \langle \theta^*, x_{\ell,i} \rangle > \mathrm{UCB}_t(\hat{r}_{t,i}),$$

or equivalently

$$\sum_{\ell \leq t \, : \, i_\ell = i} \big( \langle \theta^*, x_{\ell,i} \rangle - r_{\ell,i} \big) > 2\sqrt{n_t(i)\log(T)}.$$

Define the *good event* $\mathcal{G}$ as the event that

$$\mathcal{G} := \big\{ \mathrm{LCB}_t(\hat{r}_{t,i}) \leq \hat{r}_{t,i}^* \leq \mathrm{UCB}_t(\hat{r}_{t,i}) \; \forall t \in [T], i \in [K] \big\}.$$

By Hoeffding's inequality, we know that the good event occurs with probability at least $\mathbb{P}(\mathcal{G}) \geq 1 - \frac{1}{T^2}$. Next, let

$$\tau_i := \min\{t \in [T] \colon i \notin A_t\}$$

denote the first round in which $i$ is no longer active and, by convention, let $\tau_i = T$ if $i \in A_T$. By design of the grim trigger condition, note that $\tau_i = T$ for all $i \in [K]$ on the good event $\mathcal{G}$ if all arms always report truthfully.

We now provide a general result bounding the maximal amount of manipulation any arm can exercise before being eliminated by GGTM.

**Lemma D.1.** *On the good event $\mathcal{G}$, for any round $t \in [T]$ and any arm $i \in A_t$ it holds that*

$$\sum_{\ell \leq t \, : \, i_\ell = i} \big( \langle \theta^*, x_{\ell,i} \rangle - x_{\ell,i}^* \big) \leq 4\sqrt{n_t(i)\log(T)}.$$

*From the definition of $\tau_i$ this entails that*

$$\sum_{\ell \leq \tau_i \, : \, i_\ell = i} \big( \langle \theta^*, x_{\ell,i} \rangle - x_{\ell,i}^* \big) \leq 4\sqrt{n_{\tau_i}(i)\log(T)}.$$

*Proof.* On the good event $\mathcal{G}$, it holds that

$$\sum_{\ell \leq t \, : \, i_\ell = i} \big( \langle \theta^*, x_{\ell,i}^* \rangle - r_{\ell,i} \big) \in \big[ -2\sqrt{n_t(i)\log(T)}, +2\sqrt{n_t(i)\log(T)} \big],$$

which implies that

$$\sum_{\ell \leq t \, : \, i_\ell = i} \big( \langle \theta^*, x_{\ell,i} \rangle - r_{\ell,i} \big) \geq \sum_{\ell \leq t \, : \, i_\ell = i} \langle \theta^*, x_{\ell,i} - x_{\ell,i}^* \rangle - 2\sqrt{n_t(i)\log(T)}.$$

Hence, if $\sum_{\ell \leq t \, : \, i_\ell = i} \langle \theta^*, x_{\ell,i} - x_{\ell,i}^* \rangle > 4\sqrt{n_t(i)\log(T)}$, then

$$\sum_{\ell \leq t \, : \, i_\ell = i} \big( \langle \theta^*, x_{\ell,i} \rangle - r_{\ell,i} \big) > 2\sqrt{n_t(i)\log(T)},$$

which means that arm $i$ is eliminated from $A_t$. Finally, $\tau_i$ is defined as the first round such that $i \notin A_t$ so that

$$\sum_{\ell \leq t \, : \, i_\ell = i} \big( \langle \theta^*, x_{\ell,i} \rangle - x_{\ell,i}^* \big) \leq 4\sqrt{n_t(i)\log(T)}$$

for all $t \leq \tau_i$ and $\sum_{\ell \leq \tau_i + 1 \, : \, i_\ell = i} \big( \langle \theta^*, x_{\ell,i} \rangle - x_{\ell,i}^* \big) > 4\sqrt{n_{\tau_i}(i)\log(T)}$. $\qquad \square$

For completeness, we also formally state the fact that on the good event $\mathcal{G}$ any truthful arm is not eliminated with high probability.

**Lemma D.2.** *If arm $i$ reports truthfully every round, i.e., plays strategy $\sigma_i^*$ with $x_{t,i} = x_{t,i}^*$ for all round $t \in [T]$, then on the good event $\mathcal{G}$ arm $i$ stays active for all rounds.*

*Proof.* When arm $i$ is truthful, then $\sum_{\ell \leq t:\ i_\ell = i} \langle \theta^*, x_{\ell,i} \rangle = \hat{r}_{t,i}^*$. On the good event, $\hat{r}_{t,i}^* \leq \mathrm{UCB}_t(\hat{r}_{t,i})$ for all $t \in [T]$. Hence, the grim trigger condition is never satisfied and arm $i$ remains active throughout all $T$ rounds. $\qquad\square$

### D.2 Proof of Theorem 4.1

*Proof of Theorem 4.1.* We have to show that the strategy profile $\boldsymbol{\sigma}^*$, where every arm always truthfully reports their context, i.e., $x_{t,i} = x_{t,i}^*$ for all $(t,i) \in [T] \times [K]$, forms a $\tilde{\mathcal{O}}(\sqrt{T})$-Nash equilibrium for the arms under GGTM. We do this by showing that any deviating strategy $\sigma_i$ for arm $i$ cannot gain more than this $\sqrt{T}$ clicks. Recall that $i_t^*$ is the optimal arm in round $t$ and $i_t$ the arm the learner selects.

We begin by deriving the minimum utility of every arm when everyone is truthful. To this end, let $n_T^*(i) := \sum_{t=1}^{T} \mathbb{1}\{i_t^* = i\}$ be the number of times arm $i$ is the optimal arm. If every arm $i$ is truthful, then on the good event $\mathcal{G}$ no arm gets eliminated (Lemma D.2) and $\langle \theta^*, x_{t,i} \rangle = \langle \theta^*, x_{t,i}^* \rangle$ for all $(t,i) \in [T] \times [K]$. As a result, GGTM pulls the optimal arm $i_t^*$ in every round $t$. First, note that:

$$\mathbb{E}_{\boldsymbol{\sigma}^*}[n_T(i)] \geq n_T^*(i) - \frac{1}{T},$$

because on the good event $\mathcal{G}$ (when everyone is truthful), we have $n_T(i) \geq n_T^*(i)$. Since by construction $\mathbb{P}(\mathcal{G}) \geq 1 - 1/T^2$, the lower bound follows.

Next, we bound the utility of a deviating strategy $\sigma_i$ in response to GGTM and the other arms' truthful strategies $\sigma_{-i}^*$. On the good event $\mathcal{G}$, when the arms play strategies $(\sigma_i, \sigma_{-i}^*)$, we have

$$
\begin{aligned}
n_T(i) &= \sum_{t=1}^{T} \mathbb{1}\{i_t = i, i_t^* = i\} + \sum_{t=1}^{T} \mathbb{1}\{i_t = i, i_t^* \neq i\} \\
&\leq \sum_{t=1}^{T} \mathbb{1}\{i_t^* = i\} + \sum_{t=1}^{T} \mathbb{1}\{i_t = i, i_t^* \neq i\} \\
&= n_T^*(i) + \sum_{t=1}^{T} \mathbb{1}\{i_t = i, i_t^* \neq i\}.
\end{aligned}
$$

We will now bound the sum on the right hand side from above.

Every arm $j \neq i$ is truthful and therefore, on the good event, $j \in A_t$ for all $t$. If the optimal arm is not $i$, i.e., $i_t^* \neq i$, it means that $\langle \theta^*, x_{t,i_t^*}^* - x_{t,i}^* \rangle > 0$. Next, since GGTM selects the arms greedily according to the reported reward, the event $i_t = i$ implies that

$$\langle \theta^*, x_{t,i} \rangle \geq \langle \theta^*, x_{t,i_t^*}^* \rangle,$$

where we used that any arm $i_t^* \neq i$ is truthful so that $\langle \theta^*, x_{t,i_t^*} \rangle = \langle \theta^*, x_{t,i_t^*}^* \rangle$. As a result, we can apply Lemma D.1 to obtain

$$
\begin{aligned}
\sum_{t=1}^{T} \mathbb{1}\{i_t^* \neq i, i_t = i\} \langle \theta^*, x_{t,i_t^*}^* - x_{t,i}^* \rangle &\leq \sum_{t=1}^{T} \mathbb{1}\{i_t = i, i_t^* \neq i\} \langle \theta^*, x_{t,i} - x_{t,i}^* \rangle \\
&\leq \sum_{t=1}^{\tau_i} \mathbb{1}\{i_t = i\} \langle \theta^*, x_{t,i} - x_{t,i}^* \rangle \\
&\leq 4\sqrt{n_{\tau_i}(i) \log(T)}.
\end{aligned}
$$

Since for $i_t^* \neq i$ the gap $\langle \theta^*, x_{t,i_t^*}^* - x_{t,i}^* \rangle$ is positive and assumed to be constant, we get that $\sum_{t=1}^{T} \mathbb{1}\{i_t^* \neq i, i_t = i\} \leq \mathcal{O}(\sqrt{n_{\tau_i}(i) \log(T)})$. We coarsely upper bound $n_{\tau_i}(i)$ by $T$ and using that the good event $\mathcal{G}$ has probability at least $1 - 1/T^2$, we obtain

$$\mathbb{E}_{\sigma_i, \sigma_{-i}^*}[n_T(i)] \leq n_T^*(i) + \mathcal{O}\left( \sqrt{T \log(T)} \right).$$

We have thus shown that

$$\mathbb{E}_{\boldsymbol{\sigma}^*}[n_T(i)] \geq \mathbb{E}_{\sigma_i, \sigma_{-i}^*}[n_T(i)] + \mathcal{O}\left( \sqrt{T \log(T)} \right)$$

for any deviating (dishonest) strategy $\sigma_i$. This means that $\boldsymbol{\sigma}^*$ is a $\tilde{\mathcal{O}}(\sqrt{T})$-Nash equilibrium for the arms.

Finally, the regret of GGTM when the arms are truthful is quickly bounded by $1/T$ by using the fact that on the good event no arm gets eliminated and, therefore, GGTM picks the round-optimal arm every round. The event that $\mathcal{G}$ does not hold has probability at most $1/T^2$ which implies expected regret $1/T$, i.e., $R_T(\text{GGTM}, \boldsymbol{\sigma}^*) \leq 1/T$.

$\square$

### D.3  Proof of Theorem 4.2

*Proof of Theorem 4.2.* The proof of Theorem 4.2 is notably more involved than that of Theorem 4.1, even though the general proof idea remains similar.

We begin by decomposing of GGTM into the rounds where the optimal arm is active and the rounds in which it is being ignored. To this end, recall the definition of the arm that is optimal in round $t$ as $i_t^* := \operatorname{argmax}_{i \in [K]} \langle \theta^*, x_{t,i}^* \rangle$. We have

$$R_T = \underbrace{\mathbb{E}\left[ \sum_{t=1}^{T} \mathbb{1}\{i_t^* \in A_t\} \langle \theta^*, x_{t,i_t^*}^* - x_{t,i_t}^* \rangle \right]}_{I_1} + \underbrace{\mathbb{E}\left[ \sum_{t=1}^{T} \mathbb{1}\{i_t^* \notin A_t\} \langle \theta^*, x_{t,i_t^*}^* - x_{t,i_t}^* \rangle \right]}_{I_2}.$$

We now bound $I_1$ and $I_2$ separately as follows.

**Lemma D.3** (Bounding $I_1$).

$$\mathbb{E}\left[ \sum_{t=1}^{T} \mathbb{1}\{i_t^* \in A_t\} \langle \theta^*, x_{t,i_t^*}^* - x_{t,i_t}^* \rangle \right] \leq \mathcal{O}\left( \sqrt{KT \log(T)} \right).$$

*Proof.* Let $i_t$ denote the selection of GGTM in round $t$. Recall that GGTM greedily selects the arm in $A_t$ with highest reported value in round $t$, that is, $\langle \theta^*, x_{t,i_t} \rangle = \max_{i \in A_t} \langle \theta^*, x_{t,i} \rangle$. Consequently, on event $\{i_t^* \in A_t\}$, we have

$$\langle \theta^*, x_{t,i_t^*}^* \rangle \leq \langle \theta^*, x_{t,i_t^*} \rangle \leq \max_{i \in A_t} \langle \theta^*, x_{t,i} \rangle = \langle \theta^*, x_{t,i_t} \rangle,$$

where the first inequality holds by the assumption the optimal arm $i_t^*$ reports their value at least as high as their true value. As a consequence, it holds that $\langle \theta^*, x_{t,i_t^*}^* - x_{t,i_t}^* \rangle \leq \langle \theta^*, x_{t,i_t} - x_{t,i_t}^* \rangle$ which implies:

$$\mathbb{E}\left[ \sum_{t=1}^{T} \mathbb{1}\{i_t^* \in A_t\} \langle \theta^*, x_{t,i_t^*}^* - x_{t,i_t}^* \rangle \right] \leq \mathbb{E}\left[ \sum_{t=1}^{T} \mathbb{1}\{i_t^* \in A_t\} \langle \theta^*, x_{t,i_t} - x_{t,i_t}^* \rangle \right]. \quad (5)$$

When $i_t^* \in A_t$, a necessary condition for arm $i$ to be selected in round $t$ (i.e., $i_t = i$) is that $t \leq \tau_i$. Finally, we split the sum into each arm's contribution and apply Lemma D.1 to obtain

$$(5) \leq \mathbb{E}\left[ \sum_{i=1}^{K} \sum_{t=1}^{\tau_i} \mathbb{1}\{i_t = i\} \langle \theta^*, x_{t,i_t} - x_{t,i_t}^* \rangle \right] \leq \mathbb{E}\left[ \sum_{i=1}^{K} 4\sqrt{n_{\tau_i}(i) \log(T)} \right] \leq 4\sqrt{KT \log(T)},$$

where the last step follows from Jensen's inequality by bounding $n_{\tau_i}(i)$ by $n_T(i)$ and using that $\sum_{i=1}^{K} n_T(i) \leq T$ by definition of $n_T(i)$. $\square$

While bounding $I_1$ is fairly straightforward and we did not have to rely on the fact that the arms respond in Nash equilibrium, bounding $I_2$ becomes more challenging as we must argue that it is in each arms' interest to maintain active for a sufficiently long time.

**Lemma D.4** (Bounding $I_2$).

$$\mathbb{E}\left[\sum_{t=1}^{T} \mathbb{1}\{i_t^* \notin A_t\}\big(\mu_{t,i_t^*}^* - \mu_{t,i_t}^*\big)\right] \leq 5K^2\sqrt{KT\log(T)} \tag{6}$$

*Proof.* To bound $I_2$ we argue via the best response property of the Nash equilibrium. This requires some intermediate steps. We begin with a lower bound on the expected number of selections any arm must receive when the arms act according to a Nash equilibrium under GGTM.

Recall the definition $n_\tau^*(i) := \sum_{t=1}^{\tau} \mathbb{1}\{i_t^* = i\}$ and that the indicator variables $\mathbb{1}\{i_t^* = i\}$ are not random, since we work under an adversarially chosen sequence of true contexts. In contrast, the indicator $\mathbb{1}\{i_t = i\}$ is a random variable as it generally depends on the random reward observations and any randomization of the algorithm.

The following lemma provides a lower bound on the number of allocations any arm must receive in equilibrium. To prove the lemma, we show that we are able to protect any truthful arm from losing more than order $\sqrt{KT}$ allocations to manipulating arms. This is crucial as it would be impossible to incentivize approximately truthful arm behavior if an arm would lose too many allocations, e.g., order $T$ many, by doing so.

A key challenge here is that under two different strategies $\sigma_i$ and $\sigma_i'$, the set of active arms can be quite different. This is the case since even though we estimate each arm's expected reward independently, arm $i$ can still slightly influence the elimination of some other arm $j$ by poaching selections from them. As a result, we must content ourselves with a more conservative bound than one may originally expect.

**Lemma D.5.** *Let $\boldsymbol{\sigma} \in \mathrm{NE}(\mathrm{GGTM})$. Then,*

$$\mathbb{E}_{\boldsymbol{\sigma}}[n_T(i)] \geq n_T^*(i) - \mathcal{O}\big(\sqrt{KT\log(T)}\big).$$

*In particular, it holds that $\mathbb{E}_{\boldsymbol{\sigma}}[n_t(i)] \geq n_t^*(i) - \mathcal{O}\big(\sqrt{KT\log(T)}\big)$ for any $t \in [T]$.*

*Proof.* We use the fact that if $\boldsymbol{\sigma} = (\sigma_1, \ldots, \sigma_K)$ is a NE under GGTM, then $\sigma_i$ must be a best response to $\sigma_{-i}$, i.e., $\mathbb{E}_{\sigma_i,\sigma_{-i}}[n_T(i)] \geq \mathbb{E}_{\sigma_i',\sigma_{-i}}[n_T(i)]$ for all strategies $\sigma_i'$. In particular, it must hold for the truthful strategy $\sigma_i^*$ that

$$\mathbb{E}_{\sigma_i,\sigma_{-i}}[n_T(i)] \geq \mathbb{E}_{\sigma_i^*,\sigma_{-i}}[n_T(i)].$$

We focus on the good event $\mathcal{G}$ so that $i \in A_t$ for all $t$ given that arm $i$ is truthful. We are interested in the number of rounds such that $i_t^* = i$ and $i_t \neq i$. Given strategies $(\sigma_i^*, \sigma_{-i})$ so that $i \in A_T$ on the good event, we have

$$\sum_{t=1}^{T} \mathbb{1}\{i_t^* = i, i_t \neq i\} = \sum_{j \neq i}\sum_{t=1}^{\tau_j} \mathbb{1}\{i_t^* = i, i_t = j\}.$$

Note that $i_t = j$ with $i_t^* = i \in A_t$ implies that $\langle \theta^*, x_{t,j}\rangle \geq \langle \theta^*, x_{t,i}\rangle$. Moreover, because $i$ is truthful and $i_t^* = i$, we have $\langle \theta^*, x_{t,i}\rangle = \langle \theta^*, x_{t,i_t^*}\rangle$ so that $\langle \theta^*, x_{t,j}\rangle > \langle \theta^*, x_{t,i_t^*}\rangle$. As a result,

$$\langle \theta^*, x_{t,i_t^*}^* - x_{t,j}^*\rangle < \langle \theta^*, x_{t,j} - x_{t,j}^*\rangle.$$

It then follows from Lemma D.1 that

$$\sum_{t=1}^{\tau_j} \mathbb{1}\{i_t^* = i, i_t = j\}\langle \theta^*, x_{t,i_t^*}^* - x_{t,j}^*\rangle < \sum_{t=1}^{\tau_j} \mathbb{1}\{i_t^* = i, i_t = j\}\langle \theta^*, x_{t,j} - x_{t,j}^*\rangle \tag{7}$$

$$\leq \sum_{t=1}^{\tau_j} \mathbb{1}\{i_t = j\}\langle \theta^*, x_{t,j} - x_{t,j}^*\rangle \tag{8}$$

$$\leq 4\sqrt{n_{\tau_j}(j)\log(T)}. \tag{9}$$

Since $\langle \theta^*, x_{t,i_t^*}^* - x_{t,j}^* \rangle$ is constant for $i_t^* = i, i_t = j$, we obtain

$$\sum_{j \neq i} \sum_{t=1}^{T} \mathbb{1}\{i_t^* = i, i_t \neq i\} \leq \sum_{j \neq i} \mathcal{O}\left(\sqrt{n_{\tau_j}(j) \log(T)}\right) \leq \mathcal{O}\left(\sqrt{KT \log(T)}\right),$$

where the last inequality follows from Jensen's inequality. Recalling that $\mathbb{P}(\mathcal{G}) \geq 1 - 1/T^2$, this provides us with the following lower bound on the utility of the truthful strategy

$$\mathbb{E}_{\sigma_i^*, \sigma_{-i}}[n_T(i)] = \mathbb{E}_{\sigma_i^*, \sigma_{-i}}\left[\sum_{t=1}^{T} \mathbb{1}\{i_t = i\}\right]$$

$$\geq \mathbb{E}_{\sigma_i^*, \sigma_{-i}}\left[\sum_{t=1}^{T} \mathbb{1}\{i_t^* = i\} - \sum_{t=1}^{T} \mathbb{1}\{i_t^* = i, i_t \neq i\}\right]$$

$$\geq n_T^*(i) - \mathcal{O}\left(\sqrt{KT \log(T)}\right)$$

Note that we, like before, we account for the event $\mathcal{G}^c$ by increasing the constant factor by one, since $(1 - 1/T^2)n_T(i) \geq n_T(i) - 1/T$ as $n_T(i) \leq T$.

Since $\sigma_i$ has to be a best response to $\sigma_{-i}$, it must be as least as good as $\sigma_i^*$ sot that

$$\mathbb{E}_{\sigma}[n_T(i)] \geq \mathbb{E}_{\sigma_i^*, \sigma_{-i}}[n_T(i)] \geq n_T^*(i) - \mathcal{O}\left(\sqrt{KT \log(T)}\right).$$

To get the result for any $t \in [T]$, suppose that on the good event $\mathcal{G}$ it holds that $n_t(i) < n_t^*(i) - \omega\left(\sqrt{KT \log(T)}\right)$. Now, recall from equation (7) that the number of rounds such that $i_t^* = i$ and $i_t \neq i$ is bounded by $\mathcal{O}(\sqrt{Kt \log(T)})$ on event $\mathcal{G}$. Hence, since we assumed that $n_t(i) < n_t^*(i) - \omega(\sqrt{KT \log(T)})$ and $\langle \theta^*, x_{t,i} - x_{t,i}^* \rangle \geq 0$, it must hold that $\tau_i < t$. Consequently, on the good event $\mathcal{G}$, we obtain

$$n_{\tau_i}(i) \leq n_t(i) < n_t^*(i) - \omega\left(\sqrt{KT \log(T)}\right).$$

This implies that

$$\mathbb{E}_{\sigma}[n_{\tau_i}(i)] < \left(1 - 1/T^2\right)\left(n_t^*(i) - \omega\left(\sqrt{KT \log(T)}\right)\right) + 1/T \leq n_t^*(i) - \omega\left(\sqrt{KT \log(T)}\right).$$

This stands in contradiction to the earlier lower bound of $\mathbb{E}_{\sigma}[n_T(i)] = \mathbb{E}_{\sigma}[n_{\tau_i}(i)] \geq n_T^*(i) - \mathcal{O}\left(\sqrt{KT \log(T)}\right)$.

$\square$

Next, we provide an upper bound on the number of times an arm is pulled in any Nash equilibrium. In other words, we bound the profit any arm can make under GGTM from misreporting contexts.

**Lemma D.6.** *Let $\sigma$ be any NE under* GGTM. *Then,*

$$\mathbb{E}_{\sigma}[n_T(i)] \leq \mathbb{E}_{\sigma}[n_{\tau_i}^*(i)] + \mathcal{O}\left((K-1)\sqrt{KT \log(T)}\right)$$

*Proof.* Note that $\sum_{i=1}^{K} n_{\tau}^*(i) = \sum_{i=1}^{K} \sum_{t=1}^{\tau} \mathbb{1}\{i_t^* = i\} = \tau$ for any $\tau \in [T]$. Using Lemma D.5, we then obtain

$$\mathbb{E}_{\sigma}[n_T(i)] = \mathbb{E}_{\sigma}[n_{\tau_i}(i)]$$

$$= \mathbb{E}_{\sigma}\left[\sum_{t=1}^{\tau_i} \mathbb{1}\{i_t = i\}\right]$$

$$= \mathbb{E}_{\sigma}\left[\sum_{t=1}^{\tau_i} (1 - \mathbb{1}\{i_t \neq i\})\right]$$

$$= \mathbb{E}_{\sigma}[\tau_i] - \sum_{j \neq i} \mathbb{E}_{\sigma}[n_{\tau_i}(j)]$$

$$\leq \mathbb{E}_{\sigma}[\tau_i] - \sum_{j \neq i} \mathbb{E}_{\sigma}[n_{\tau_i}^*(j)] + \mathcal{O}\left((K-1)\sqrt{KT \log(T)}\right)$$

$$= \mathbb{E}_{\sigma}[n_{\tau_i}^*(i)] + \mathcal{O}\left((K-1)\sqrt{KT \log(T)}\right)$$

$\square$

Combining Lemma D.5 and Lemma D.6 we get for any Nash equilibrium $\boldsymbol{\sigma} \in \text{NE(GGTM)}$ that

$$\mathbb{E}_{\boldsymbol{\sigma}}[n_T^*(i)] - \mathcal{O}\big(\sqrt{KT\log(T)}\big) \leq \mathbb{E}_{\boldsymbol{\sigma}}[n_T(i)] \leq \mathbb{E}_{\boldsymbol{\sigma}}[n_{\tau_i}^*(i)] - \mathcal{O}\big((K-1)\sqrt{KT\log(T)}\big),$$

which implies that

$$\mathbb{E}_{\boldsymbol{\sigma}}[n_T^*(i) - n_{\tau_i}^*(i)] \leq \mathcal{O}\big(K\sqrt{KT\log(T)}\big). \tag{10}$$

The expression $n_T^*(i) - n_{\tau_i}^*(i)$ is the number of rounds where arm $i$ was optimal but already eliminated by the grim trigger. As a result, we can express the total number of rounds where the round-optimal arm $i_t^*$ was no longer active as follows.

**Lemma D.7.** *For any $\boldsymbol{\sigma}$, we have*

$$\mathbb{E}_{\boldsymbol{\sigma}}\left[\sum_{t=1}^T \mathbb{1}\{i_t^* \notin A_t\}\right] = \sum_{i=1}^K \mathbb{E}_{\boldsymbol{\sigma}}[n_T^*(i) - n_{\tau_i}^*(i)].$$

*Proof.* Rewriting $\{i_t^* \notin A_t\}$ yields

$$\begin{aligned}
\mathbb{E}_{\boldsymbol{\sigma}}\left[\sum_{t=1}^T \mathbb{1}\{i_t^* \notin A_t\}\right] &= \sum_{i=1}^K \mathbb{E}_{\boldsymbol{\sigma}}\left[\sum_{t=1}^T \mathbb{1}\{i_t^* = i, i \notin A_t\}\right] \\
&= \sum_{i=1}^K \mathbb{E}_{\boldsymbol{\sigma}}\left[\sum_{t=1}^T \mathbb{1}\{i_t^* = i\} - \sum_{t=1}^T \mathbb{1}\{i_t^* = i, i \in A_t\}\right] \\
&= \sum_{i=1}^K \mathbb{E}_{\boldsymbol{\sigma}}\left[n_T^*(i) - n_{\tau_i}^*(i)\right].
\end{aligned}$$

$\square$

Finally, note that $\langle \theta^*, x_{t,i_t^*}^* - x_{t,i_t}^* \rangle \leq 1$ so that from equation (10) it follows that

$$\begin{aligned}
\mathbb{E}\left[\sum_{t=1}^T \mathbb{1}\{i_t^* \notin A_t\}\langle \theta^*, x_{t,i_t^*}^* - x_{t,i_t}^* \rangle\right] &\leq \mathbb{E}\left[\sum_{t=1}^T \mathbb{1}\{i_t^* \notin A_t\}\right] \\
&= \sum_{i=1}^K \mathbb{E}\left[n_T^*(i) - n_{\tau_i}^*(i)\right] \\
&\leq \sum_{i=1}^K \mathcal{O}\big(K\sqrt{KT\log(T)}\big) = \mathcal{O}\big(K^2\sqrt{KT\log(T)}\big).
\end{aligned}$$

$\square$

Connecting the bound on $I_1$ and $I_2$, we then obtain the final regret bound of Theorem 4.2

$$R_T(\text{GGTM}, \boldsymbol{\sigma}) \leq \mathcal{O}\left(\underbrace{\sqrt{KT\log(T)}}_{\text{Lemma D.3}} + \underbrace{K^2\sqrt{KT\log(T)}}_{\text{Lemma D.4}}\right) \leq \tilde{\mathcal{O}}\left(K^2\sqrt{KT}\right).$$

$\square$

**Remark D.8.** *We want to briefly comment on the existence of a Nash equilibrium. Since each arm's strategy space, given by $\{x \in \mathbb{R}^d : \|x\|_2 \leq 1\}$ in every round, is continuous, it is not obvious that a Nash equilibrium for the arms exists under every algorithm. However, Glickberg's theorem shows that the continuity of the arms' utility in the arms' strategies is a sufficient condition for the existence of a NE, since the strategy space is compact. We can then ensure the continuity by, e.g., choosing arms proportionally to $\exp(T\langle \theta^*, x_{t,i}\rangle)$ in GGTM and $\exp(T\text{UCB}_t(x_{t,i}))$ in OptGTM, and remarking that the probability of eliminating arm $i$ in round $t$ is continuous in $x_{t,i}$ conditional on any history. Due to the exponential scaling with $T$ the effect of such slight randomization is negligible in the regret analysis.*

# E   Proof of Theorem 5.1 and Theorem 5.2

The following preliminaries are fundamental to the proofs of Theorem 5.1 and Theorem 5.2 so that we derive them jointly here.

## E.1   Preliminaries

We begin by recalling the definition of the least-squares estimator w.r.t. arm $i$'s reported contexts and the corresponding confidence ellipsoid $C_{t,i}$. Note that since the arms are manipulating their contexts, the least-squares estimator may not accurately estimate $\theta^*$ and $\theta^*$ may not be contained in $C_{t,i}$ w.h.p. As discussed in the main text, since accurate estimation of $\theta^*$ appears hopeless, the main idea idea is to incentivize arms to report contexts such that our expected reward does not differ substantially from the observed reward.

The least-squares estimator w.r.t. arm $i$ is given by

$$\hat{\theta}_{t,i} = \underset{\theta \in \mathbb{R}^d}{\operatorname{argmin}} \left( \sum\nolimits_{\ell < t:\, i_\ell = i} \left( \langle \theta, x_{\ell,i} \rangle - r_{\ell,i} \right)^2 + \lambda \|\theta\|_2^2 \right),$$

where $\lambda > 0$. In the algorithm, we set the penalty factor to $\lambda = 1$. The closed form solution is then given by

$$\hat{\theta}_{t,i} = V_{t,i}^{-1} \sum_{\ell < t:\, i_\ell = i} x_{\ell,i} r_{\ell,i} \quad \text{with} \quad V_{t,i} = \lambda I + \sum_{\ell < t:\, i_\ell = i} x_{\ell,i} x_{\ell,i}^\top.$$

The confidence set $C_{t,i}$ is defined as

$$C_{t,i} = \left\{ \theta \in \mathbb{R}^d \colon \|\hat{\theta}_{t,i} - \theta\|_{V_t}^2 \le \beta_{t,i} \right\},$$

where we let $\beta_{t,i} = \sqrt{d \log \left( \frac{1 + n_t(i)/\lambda}{\delta} \right)} + \sqrt{\lambda} S$ with $\|\theta^*\|_2 \le S$ and $\delta = 1/T^2$.

We now translate the standard result used to assert the validity of the confidence set to our situation. Clearly, when the sequence of $x_{t,i}$ differs significantly from the true contexts $x_{t,i}^*$, the true parameter $\theta^*$ will not be contained in $C_{t,i}$. Instead, we will formulate the concentration result as follows.

**Lemma E.1.** *Suppose there exists $\theta_i^* \in \mathbb{R}^d$ such that for all $t$ with $i_t = i$:*

$$\langle \theta^*, x_{t,i}^* \rangle = \langle \theta_i^*, x_{t,i} \rangle. \tag{11}$$

*In other words, the reported features $x_{t,i}$ are linearly realizable by some parameter $\theta_i^*$.*

*For any $\delta \in (0,1)$ let the confidence size be*

$$\beta_{t,i} = \sqrt{d \log \left( \frac{1 + n_t(i)/\lambda}{\delta} \right)} + \sqrt{\lambda} S,$$

*where $\|\theta_i^*\|_2 \le S$. Note that the typical expression also includes some constant $L$ such that $\|x_{t,i}\|_2 \le L$, which we here simply set to 1. With probability at least $1 - \delta$ it then holds that $\theta_i^* \in C_{t,i}$. In what follows, we choose $\delta = 1/T^2$.*

*As a special case, when arm $i$ is always truthful so that $x_{t,i} = x_{t,i}^*$, the true parameter $\theta^*$ trivially satisfies (11) and the result reduces to the standard confidence bound statement [1, 24] restricted to observations from arm $i$.*

*Proof.* Let $\theta_i^*$ satisfy (11). Then, note that $r_{t,i} \coloneqq \langle \theta^*, x_{t,i}^* \rangle + \eta_t = \langle \theta_i^*, x_{t,i} \rangle + \eta_t$. Hence, the sequence of reported features $x_{t,i}$ and observed reward $r_{t,i}$ yield a standard linear contextual bandit structure with unknown parameter $\theta_i^*$ (instead of $\theta^*$). Then, to obtain the confidence bound follow the arguments from [1, 24], where we remark that we choose the confidence radius $\beta_{t,i}$ arm specific. However, we could also choose a larger confidence radius such as $\beta_t \approx d \log(t)$ or even constant $\beta \approx d \log(T)$. This will only have a negligible effect on the final regret. $\square$

The grim trigger condition (4) of OptGTM stated that arm $i$ gets eliminated in round $t$ if

$$\sum_{\ell \le t:\, i_\ell = i} \left( \langle \hat{\theta}_{\ell,i}, x_{\ell,i} \rangle - \sqrt{\beta_\ell} \|x_{\ell,i}\|_{V_{\ell,i}^{-1}} \right) > \sum_{\ell \le t:\, i_\ell = i} r_{\ell,i} + 2\sqrt{n_t(i) \log(T)}. \tag{12}$$

Equivalently, $\sum_{\ell \leq t:\ i_\ell = i} \mathrm{LCB}_{\ell,i}(x_{\ell,i}) > \mathrm{UCB}_t(\hat{r}_{t,i})$.

As a sanity check, we show that when an arm always reports truthfully, i.e., $x_{t,i} = x^*_{t,i}$ for all $t$, it doesn't get eliminated with probability at least $1 - 1/T^2$.

**Lemma E.2.** *When arm $i$ always reports truthfully it does not get eliminated with high probability, that is, $i \in A_T$ with probability at least $1 - 1/T^2$.*

*Proof.* We consider the event that the true parameter $\theta^*$ is contained in $C_{t,i}$, i.e., $\mathcal{G}'_i := \{\theta^* \in C_{t,i} \,\forall t \in [T]\}$. The event $\mathcal{G}'_i$ has probability at least $1 - 1/T^2$ according to Lemma E.1 when arm $i$ is truthful. Moreover, suppose that the reward observations concentrate as well, i.e., we assume the good event $\mathcal{G} := \{\mathrm{LCB}_t(\hat{r}_{t,i}) \leq \hat{r}^*_{t,i} \leq \mathrm{UCB}_t(\hat{r}_{t,i}) \,\forall t \in [T], i \in [K]\}$. Recall that $\mathcal{G}$ has probability at least $1 - 1/T^2$ according to Hoeffding's inequality. A union bound then shows that the intersection of the two events has probability at least $1 - 2/T^2$.

Now, since arm $i$ is truthful, we have $x_{t,i} = x^*_{t,i}$ and $\langle \theta, x_{t,i} \rangle = \langle \theta, x^*_{t,i} \rangle$ for all $\theta \in \mathbb{R}^d$. Using Cauchy-Schwarz inequality and the fact that $\theta^* \in C_{t,i}$, we get that

$$\langle \hat{\theta}_{t,i}, x_{t,i} \rangle - \langle \theta^*, x^*_{t,i} \rangle = \langle \hat{\theta}_{t,i} - \theta^*, x^*_{t,i} \rangle \leq \|\hat{\theta}_{t,i} - \theta^*\|_{V_{t,i}} \|x^*_{t,i}\|_{V_{t,i}^{-1}} \leq \sqrt{\beta_{t,i}} \|x_{t,i}\|_{V_{t,i}^{-1}}$$

Moreover, as we work on the good event $\mathcal{G}$, we have

$$\sum_{\ell \leq t:\ i_\ell = i} \left( \langle \theta^*, x^*_{\ell,i} \rangle - r_{\ell,i} \right) \in [-2\sqrt{n_t(i)\log(T)}, +2\sqrt{n_t(i)\log(T)}].$$

Combining these two statements yields

$$\sum_{\ell \leq t:\ i_\ell = i} \left( \langle \hat{\theta}_{\ell,i}, x_{t,i} \rangle - r_{\ell,i} \right) \leq \sum_{\ell \leq t:\ i_\ell = i} \sqrt{\beta_{\ell,i}} \|x_{\ell,i}\|_{V_{\ell,i}^{-1}} + 2\sqrt{n_t(i)\log(T)}$$

for all $t \in [T]$. In other words, the grim trigger condition is never satisfied so that $i \in A_T$ on event $\mathcal{G} \cap \mathcal{G}'_i$, which, as we saw, occurs with probability at least $1 - 1/T^2$. $\qquad\square$

We now analyze the grim trigger of the OptGTM algorithm. As before, let $\tau_i := \min\{t: i \notin A_t\}$ with the convention that $\tau_i = T$ if $i \in A_T$. The following lemma upper bounds the total amount of manipulation that an arm can exert before being eliminated by OptGTM's grim trigger elimination rule.

**Lemma E.3.** *On the good event $\mathcal{G}$:*

$$\sum_{t \leq \tau_i:\ i_t = i} \left( \langle \hat{\theta}_{t,i}, x_{t,i} \rangle - \langle \theta^*, x^*_{t,i} \rangle \right) \leq \sum_{t \leq \tau_i:\ i_t = i} \sqrt{\beta_{t,i}} \|x_{t,i}\|_{V_{t,i}^{-1}} + 4\sqrt{n_{\tau_i}(i)\log(T)}.$$

*Or equivalently, since* $\mathrm{UCB}_{t,i}(x_{t,i}) = \langle \hat{\theta}_{t,i}, x_{t,i} \rangle + \sqrt{\beta_{t,i}} \|x_{t,i}\|_{V_{t,i}^{-1}}$, *it holds that*

$$\sum_{t \leq \tau_i:\ i_t = i} \left( \mathrm{UCB}_{t,i}(x_{t,i}) - \langle \theta^*, x^*_{t,i} \rangle \right) \leq \sum_{t \leq \tau_i:\ i_t = i} 2\sqrt{\beta_{t,i}} \|x_{t,i}\|_{V_{t,i}^{-1}} + 4\sqrt{n_{\tau_i}(i)\log(T)}.$$

*Proof.* Let $t \in [T]$. On the good event $\mathcal{G} := \{\mathrm{LCB}_t(\hat{r}_{t,i}) \leq \hat{r}^*_{t,i} \leq \mathrm{UCB}_t(\hat{r}_{t,i}) \,\forall t \in [T], i \in [K]\}$ and by definition of $\mathrm{UCB}_{\ell,i}(x_{\ell,i}$, it follows that

$$\sum_{\ell \leq t:\ i_\ell = i} \left( \langle \hat{\theta}_{\ell,i}, x_{\ell,i} \rangle - r_{\ell,i} \right)$$

$$\geq \sum_{\ell \leq t:\ i_\ell = i} \left( \mathrm{UCB}_{\ell,i}(x_{\ell,i}) - \langle \theta^*, x^*_{\ell,i} \rangle \right) - \underbrace{\sum_{\ell \leq t:\ i_\ell = i} \sqrt{\beta_{\ell,i}} \|x_{\ell,i}\|_{V_{\ell-1,i}^{-1}} - 2\sqrt{n_t(i)\log(T)}}_{R :=}.$$

Hence, if $\sum_{\ell \leq t:\ i_\ell = i} \left( \mathrm{UCB}_{\ell,i}(x_{\ell,i}) - \langle \theta^*, x^*_{\ell,i} \rangle \right) > 2R$, then

$$\sum_{\ell \leq t:\ i_\ell = i} \left( \langle \hat{\theta}_{\ell,i}, x_{\ell,i} \rangle - r_{\ell,i} \right) > \sum_{\ell \leq t:\ i_t = i} \sqrt{\beta_{\ell,i}} \|x_{\ell,i}\|_{V_{\ell,i}^{-1}} + 2\sqrt{n_t(i)\log(T)},$$

which means that arm $i$ must have been eliminated in a previous round or in round $t$, i.e., $\tau_i > t$. Hence, for any $t \leq \tau_i$, the the left hand side must be smaller or equal to the right hand side.

Interestingly, notice that we worked on the good event $\mathcal{G}$ that only concerns the realization of the rewards and not the validity of the confidence set. This is important, since it is generally not true that the true parameter $\theta^*$ is contained in the confidence set $C_{t,i}$.

$\square$

Lastly, before we begin with the proof Theorem 5.1 and Theorem 5.2, we establish a bound on the total exploration bonus, which after some additional work follows from the well-known elliptical potential lemma [1, 24].

**Lemma E.4.** *It holds that*

$$\sum_{t \leq \tau_i : \, i_t = i} \sqrt{\beta_t} \|x_{t,i}\|_{V_{t,i}^{-1}} \leq \mathcal{O}\left(d \log(T) \sqrt{n_{\tau_i}(i)}\right).$$

*The constant on the right hand side can be derived from the choice of $\beta_{t,i}$.*

*Proof.* Before we can apply the elliptical potential lemma[1], we need to make sure that the exploration bonus does not blow up in early rounds. To this end, recall the definition of

$$V_{t,i} := \lambda I + \sum_{\ell < t : \, i_\ell = i} x_{\ell,i} x_{\ell,i}^\top.$$

Let $A = \sum_{\ell \leq t : \, i_\ell = i} x_{\ell,i} x_{\ell,i}^\top$. Note that $A$ is positive semi-definite so that $\lambda I + A$ is positive definite and its inverse $(\lambda I + A)^{-1}$ as well. The matrix inversion lemma let's us express this inverse as

$$(\lambda I + A)^{-1} = \lambda I - (\lambda I + A)^{-1} A.$$

Now, the eigenvalues of $B = (\lambda I + A)^{-1} A$ are given by $\lambda_i/(1 + \lambda_i)$, where $\lambda_i \geq 0$ are the eigenvalues of $A$, which means that $B$ is positive semi-definite. Consequently,

$$\|x_{t,i}\|_{V_{t,i}^{-1}}^2 = x_{t,i}^\top (\lambda I + A)^{-1} x_{t,i} = x_{t,i}^\top \lambda x_{t,i} - x_{t,i}^\top B x_{t,i} \leq \lambda \|x_{t,i}\|_2^2.$$

We assumed that $\|x_{t,i}\|_2^2 \leq 1$ (similarly we could assume an upper bound $L$) so that $\|x_{t,i}\|_{V_{t,i}^{-1}} \leq \sqrt{\lambda}$. For convenience, we set the penalty factor to $\lambda = 1$. We then apply Cauchy-Schwarz to get

$$\sum_{t \leq \tau_i : \, i_t = i} \sqrt{\beta_{t,i}} \|x_{t,i}\|_{V_{t,i}^{-1}} \leq \sqrt{n_{\tau_i}(i) \beta_T \sum_{t \leq \tau_i : \, i_t = i} \min\{1, \|x_{t,i}\|_{V_{t,i}^{-1}}^2\}}.$$

The elliptical potential lemma [1, 24] bounds the sum on the right hand side as

$$\sum_{t \leq \tau_i : \, i_t = i} \min\{1, \|x_{t,i}\|_{V_{t,i}^{-1}}^2\} \leq 2d \log\left(\frac{d + n_{\tau_i}(i)}{d}\right)$$

Finally, recall that we chose $\beta_{t,i} = \mathcal{O}\left(d \log\left(n_t(i)\right)\right)$, which then yields the claimed bound. $\square$

### E.2 Proof of Theorem 5.1

*Proof of Theorem 5.1.* We begin by proving that being truthful is an approximate Nash equilibrium under OptGTM.

**Truthfulness is a $\tilde{\mathcal{O}}(d\sqrt{KT})$-NE.** In a fist step, we show that if every arm is truthful, every arm is guaranteed at least $n_T^*(i) - \tilde{\mathcal{O}}(d\sqrt{T})$ utility, where $n_T^*(i) := \sum_{t=1}^{T} \mathbb{1}\{i_t^* = i\}$. To this end, we write

$$n_T(i) = \sum_{t=1}^{T} \mathbb{1}\{i_t = i, i_t^* = i\} + \sum_{t=1}^{T} \mathbb{1}\{i_t = i, i_t^* \neq i\}$$

$$\geq \sum_{t=1}^{T} \mathbb{1}\{i_t^* = i\} - \sum_{t=1}^{T} \mathbb{1}\{i_t^* = i, i_t \neq i\},$$

and our task will be bounding the sum on the right hand side. We focus on the event that $\theta^* \in C_{t,i}$ for all $(t,i) \in [T] \times [K]$, which according to Lemma E.1 occurs with probability at least $1 - 1/T^2$.

Since $\theta^* \in C_{t,i}$, we have $\mathrm{UCB}_{t,i}(x_{t,i_t}^*) \geq \langle \theta^*, x_{t,i_t^*}^* \rangle$ for every $i \in [K]$. Let $i_t^* = i$ but $i_t = j$ with $j \neq i$. Keeping in mind that $x_{t,i} = x_{t,i}^*$ for all $(t,i) \in [T] \times [K]$, since all arms are truthful, this implies that

$$\mathrm{UCB}_{t,i}(x_{t,j}^*) \geq \mathrm{UCB}_{t,i}(x_{t,i}^*) \geq \langle \theta^*, x_{t,i_t^*}^* \rangle.$$

As a result, it holds that

$$\mathrm{UCB}_{t,i}(x_{t,j}^*) - \langle \theta^*, x_{t,j}^* \rangle \geq \langle \theta^*, x_{t,i_t^*}^* - x_{t,j}^* \rangle.$$

Next, since $\theta^* \in C_{t,i}$ and $\hat{\theta}_{t,i} \in C_{t,i}$, we find that

$$\mathrm{UCB}_{t,j}(x_{t,j}^*) - \langle \theta^*, x_{t,j}^* \rangle \leq \langle \hat{\theta}_{t,j}, x_{t,j}^* \rangle - \langle \theta^*, x_{t,j}^* \rangle + \sqrt{\beta_{t,j}} \|x_{t,j}\|_{V_{t,j}^{-1}}$$

$$\leq \|\hat{\theta}_{t,j} - \theta^*\|_{V_{t,j}} \|x_{t,j}\|_{V_{t,j}^{-1}}$$

$$\leq \sqrt{\beta_{t,j}} \|x_{t,j}\|_{V_{t,j}^{-1}},$$

where the second line follows from Cauchy-Schwarz inequality. Using Lemma E.4, this implies

$$\sum_{t \leq T : i_t = j} \mathrm{UCB}_{t,j}(x_{t,j}^*) - \langle \theta^*, x_{t,j}^* \rangle \leq \sum_{t \leq T : i_t = j} \sqrt{\beta_{t,j}} \|x_{t,j}\|_{V_{t,j}^{-1}}$$

$$\leq \mathcal{O}\left( d \log(T) \sqrt{n_{\tau_j}(j)} \right).$$

Since $\langle \theta^*, x_{t,i_t^*}^* - x_{t,j}^* \rangle$ is constant for $i_t^* \neq j$, this means that

$$\sum_{t=1}^{T} \mathbb{1}\{i_t^* = i, i_t = j\} \leq \mathcal{O}\left( d \log(T) \sqrt{n_{\tau_j}(j)} \right)$$

so that by Jensen's inequality

$$\sum_{t=1}^{T} \mathbb{1}\{i_t^* = i, i_t \neq i\} = \sum_{j \neq i} \sum_{t=1}^{T} \mathbb{1}\{i_t^* = i, i_t = j\} \leq \mathcal{O}\left( d \log(T) \sqrt{KT} \right).$$

In a second step, we show that for any deviating strategy $\sigma_i$ that is not truthful, the utility of arm $i$ is upper bounded by $n_T^*(i) + \tilde{\mathcal{O}}(d\sqrt{T})$. In what follows, we work on the event that $j \in A_t$ and $\theta^* \in C_{t,j}$ for all $t \in [T]$ and $j \neq i$ and recall that this event has probability at least $1 - 1/T^2$ since the arms are reporting truthfully (see Lemma E.2). Since $j \in A_T$ for all $j \neq i$, we have

$$n_T(i) \leq \sum_{t=1}^{T} \mathbb{1}\{i_t^* = i\} + \sum_{t=1}^{T} \mathbb{1}\{i_t = i, i_t^* \neq i\},$$

and we are tasked with bounding the sum on the right hand side appropriately. Now, similarly to before if $i_t = i$ and $i_t^* \neq i$ (given $i_t^* \in A_t$), then

$$\mathrm{UCB}_{t,i}(x_{t,i}) \geq \mathrm{UCB}_{t,i_t^*}(x_{t,i_t^*}) \geq \langle \theta^*, x_{t,i_t^*}^* \rangle,$$

where we used that $\theta^* \in C_{t,i_t^*}$ since the arm $i_t^* \neq i$ is truthful. Consequently,

$$\mathrm{UCB}_{t,i}(x_{t,i}) - \langle \theta^*, x_{t,i}^* \rangle \geq \langle \theta^*, x_{t,i_t^*}^* - x_{t,i}^* \rangle.$$

Combining Lemma E.3 and Lemma E.4 tells us that

$$\sum_{t \leq \tau_i : i_t = i} \mathrm{UCB}_{t,i}(x_{t,i}) - \langle \theta^*, x_{t,i}^* \rangle \leq \mathcal{O}\left( d \log(T) \sqrt{T} \right),$$

where we coarsely upper bounded $n_{\tau_i}(i)$ by $T$. Since $\langle \theta^*, x_{t,i_t^*}^* - x_{t,i}^* \rangle$ for $i_t^* \neq i$ is positive and constant, it follows that

$$\sum_{t=1}^{T} \mathbb{1}\{i_t = i, i_t^* \neq i\} \leq \mathcal{O}\left( d \log(T) \sqrt{T} \right).$$

In summary, we have shown that

$$\mathbb{E}_{\boldsymbol{\sigma}^*}[n_T(i)] \geq n_T^*(i) - \tilde{\mathcal{O}}\left( d\sqrt{KT} \right) \quad \text{and} \quad \mathbb{E}_{\sigma_i, \sigma_{-i}^*}[n_T(i)] \leq n_T^*(i) + \tilde{\mathcal{O}}\left( d\sqrt{T} \right)$$

for any deviating strategy $\sigma_i$. Hence, $\boldsymbol{\sigma}^*$ is a $\tilde{\mathcal{O}}(d\sqrt{KT})$-Nash equilibrium under OptGTM.

**Regret analysis.** Since we maintain estimates and confidence sets for each arm independently, it is natural to decompose the regret as

$$\sum_{t=1}^{T} \langle \theta^*, x^*_{t,i^*_t} - x^*_{t,i_t} \rangle = \sum_{i=1}^{K} \sum_{t \leq T : i_t = i} \langle \theta^*, x^*_{t,i^*_t} - x^*_{t,i} \rangle. \tag{13}$$

Note that w.h.p. no truthful arm gets eliminated and $\theta^* \in C_{t,i}$ for all $(t,i) \in [T] \times [K]$. The regret analysis then proceeds similarly to that of LinUCB.

Since $\theta^* \in C_{t,i}$, we know that for any round such that $i_t = i$ that

$$\langle \theta^*, x^*_{t,i^*_t} \rangle \leq \text{UCB}_{t,i^*_t}(x^*_{t,i^*_t}) = \text{UCB}_{t,i^*_t}(x_{t,i^*_t}) \leq \text{UCB}_{t,i}(x_{t,i}) = \text{UCB}_{t,i}(x^*_{t,i}).$$

Then, again for any round with $i_t = i$, applying Cauchy-Schwarz inequality yields

$$\langle \theta^*, x^*_{t,i^*_t} - x^*_{t,i} \rangle \leq \text{UCB}_{t,i}(x^*_{t,i}) - \langle \theta^*, x^*_{t,i} \rangle \leq 2\sqrt{\beta_{t,i}} \|x_{t,i}\|_{V^{-1}_{t,i}}. \tag{14}$$

Next, first using Cauchy-Schwarz inequality and the elliptical potential lemma (Lemma E.4), and then Jensen's inequality, it follows that

$$2 \sum_{i=1}^{K} \sum_{t \leq T : i_t = i} \sqrt{\beta_{t,i}} \|x_{t,i}\|_{V^{-1}_{t,i}} \leq \mathcal{O}\left( d \log(T) \sqrt{KT} \right). \tag{15}$$

Hence, connecting equations (13)-(15), we obtain $R_T(\text{OptGTM}, \boldsymbol{\sigma}^*) \leq \tilde{\mathcal{O}}(d\sqrt{KT})$. We see that the additional $\sqrt{K}$ factor emerges due to OptGTM maintaining independent estimates for each arm. Usually a dependence on the action set size can be prevented since observations from one arm can be used for another arm as well. However, to prevent collusion in the strategic linear contextual bandit it is important to limit the influence an arm has on the selection (and elimination) of other arms. □

### E.3 Proof of Theorem 5.2

*Proof of Theorem 5.2.* We begin the proof of Theorem 5.2 by decomposing the regret into two expression, which we then separately bound.

**Decomposing strategic regret.** We now decompose the regret of OptGTM into the rounds $t$ where the optimal arm in round $t$ is still active and the rounds where it is not. Like before, let $i^*_t := \text{argmax}_{i \in [K]} \langle \theta^*, x^*_{t,i} \rangle$ be the optimal arm in round $t$. For any $\boldsymbol{\sigma} \in \text{NE}(\text{OptGTM})$, we have

$$R_T(\boldsymbol{\sigma}) = \underbrace{\mathbb{E}_{\boldsymbol{\sigma}} \left[ \sum_{t=1}^{T} \mathbb{1}\{i^*_t \in A_t\} \langle \theta^*, x^*_{t,i^*_t} - x^*_{t,i_t} \rangle \right]}_{J_1} + \underbrace{\mathbb{E}_{\boldsymbol{\sigma}} \left[ \sum_{t=1}^{T} \mathbb{1}\{i^*_t \notin A_t\} \langle \theta^*, x^*_{t,i^*_t} - x^*_{t,i_t} \rangle \right]}_{J_2}. \tag{16}$$

With the help of Lemma E.3 and Lemma E.4 we now bound $J_1$.

**Lemma E.5** (Bounding $J_1$)**.**

$$\mathbb{E}_{\boldsymbol{\sigma}} \left[ \sum_{t=1}^{T} \mathbb{1}\{i^*_t \in A_t\} \langle \theta^*, x^*_{t,i^*_t} - x^*_{t,i_t} \rangle \right] \leq \mathcal{O}\left( d\sqrt{KT} \log(T) \right).$$

*Proof.* By design of OptGTM, we we have $\langle \theta^*, x^*_{t,i^*_t} \rangle \leq \text{UCB}_{t,i^*_t}(x_{t,i^*_t}) \leq \text{UCB}_{t,i_t}(x_{t,i_t})$. Then, on the good event $\mathcal{G}$, Lemma E.3 yields

$$\sum_{t \leq \tau_i : i_t = i} \langle \theta^*, x^*_{t,i^*_t} - x^*_{t,i} \rangle \leq \sum_{t \leq \tau_i : i_t = i} \left( \text{UCB}_{t,i_t}(x_{t,i_t}) - \langle \theta^*, x^*_{t,i} \rangle \right)$$

$$\leq 2 \left( 2\sqrt{n_{\tau_i}(i) \log(T)} + \sum_{t \leq \tau_i : i_t = i} \sqrt{\beta_{t,i}} \|x_{t,i}\|_{V^{-1}_{t,i}} \right).$$

Then, on the good event, we have

$$\sum_{t=1}^{T} \mathbb{1}\{i_t^* \in A_t\}\langle \theta^*, x_{t,i_t^*}^* - x_{t,i_t}^* \rangle = \sum_{i=1}^{K} \sum_{t=1}^{\tau_i} \mathbb{1}\{i_t = i\}\langle \theta^*, x_{t,i_t^*}^* - x_{t,i}^* \rangle$$

$$\leq \sum_{i=1}^{K} \sum_{t \leq \tau_i \,:\, i_t = i} 2\sqrt{\beta_{t,i}} \|x_{t,i}\|_{V_{t,i}^{-1}} + \sum_{i=1}^{K} 2\sqrt{n_{\tau_i}(i)\log(T)}$$

$$\leq \sum_{i=1}^{K} \mathcal{O}\left(d\log(T)\sqrt{n_{\tau_i}(i)}\right) + \sum_{i=1}^{K} 2\sqrt{n_{\tau_i}(i)\log(T)}$$

$$\leq \mathcal{O}\left(d\log(T)\sqrt{KT}\right)$$

where we applied Jensen's inequality in the last inequality and used that $\sum_{i=1}^{K} n_{\tau_i}(i) \leq T$.

$\square$

**Lemma E.6** (Bounding $J_2$)**.**

$$\mathbb{E}_{\boldsymbol{\sigma}}\left[\sum_{t=1}^{T} \mathbb{1}\{i_t^* \notin A_t\}\langle \theta^*, x_{t,i_t^*}^* - x_{t,i_t}^* \rangle\right] \leq \mathcal{O}\left(dK^2\sqrt{KT}\log(T)\right).$$

*Proof.* The proof idea is the same as the one for Lemma D.4, which was used to show the regret upper bound of the Greedy Grim Trigger Mechanism (Theorem 4.2, Appendix D). Recall that by assumption $\langle \theta^*, x_{t,i_t^*}^* - x_{t,i_t}^* \rangle \leq 1$. We reuse Lemma D.7 from the proof of Theorem 4.2 to get

$$\mathbb{E}_{\boldsymbol{\sigma}}\left[\sum_{t=1}^{T} \mathbb{1}\{i_t^* \notin A_t\}\langle \theta^*, x_{t,i_t^*}^* - x_{t,i_t}^* \rangle\right] \leq \mathbb{E}_{\boldsymbol{\sigma}}\left[\sum_{t=1}^{T} \mathbb{1}\{i_t^* \notin A_t\}\right] = \sum_{i=1}^{K} \mathbb{E}_{\boldsymbol{\sigma}}\left[n_T^*(i) - n_{\tau_i}^*(i)\right],$$

where $n_t^*(i) := \sum_{s=1}^{t} \mathbb{1}\{i_s^* = i\}$ is the number of rounds up to round $t$ that $i$ is the optimal. To bound the right hand side, we first prove a lower bound on $\mathbb{E}_{\boldsymbol{\sigma}}[n_T(i)]$ for any NE $\boldsymbol{\sigma} \in \mathrm{NE}(\mathrm{OptGTM})$.

**Lemma E.7.** *Let $\boldsymbol{\sigma} \in \mathrm{NE}(\mathrm{OptGTM})$. It holds that*

$$\mathbb{E}_{\boldsymbol{\sigma}}\left[n_T(i)\right] \geq n_T^*(i) - \mathcal{O}\left(d\log(T)\sqrt{KT}\right).$$

*In particular, it holds that $\mathbb{E}_{\boldsymbol{\sigma}}\left[n_t(i)\right] \geq n_t^*(i) - \mathcal{O}\left(d\log(T)\sqrt{KT}\right)$ for $t \in [T]$.*

Since $\mathcal{G}$ occurs with probability $1 - 1/T^2$ and $n_T(i) \leq T$ by definition, on event $\mathcal{G}$, we have

$$n_T(i) \geq n_T^*(i) - \mathcal{O}\left(d\log(T)\sqrt{KT}\right) \qquad n_t(i) \geq n_t^*(i) - \mathcal{O}\left(d\log(T)\sqrt{KT}\right).$$

*Proof.* Given that arm $i$ always reports truthfully, i.e., $x_{t,i} = x_{t,i}^*$ for all $t$, consider the event that $\theta^* \in C_{t,i}$ for all $t$. Recall that this event has probability at least $1 - 1/T^2$ according to Lemma E.1.

We use the best response property of the Nash equilibrium by comparing against the truthful strategy. To this end, consider the strategy profile $(\sigma_i^*, \sigma_{-i})$ and the event that $\theta^* \in C_{t,i}$ for all $t$ as well as $\mathcal{G}$. We then have that

$$n_T(i) \geq \sum_{t=1}^{T} \mathbb{1}\{i_t^* = i\} - \sum_{t=1}^{T} \mathbb{1}\{i_t^* = i, i_t \neq i\}$$

$$= n_T^*(i) - \sum_{j \neq i} \sum_{t=1}^{\tau_i} \mathbb{1}\{i_t^* = i, i_t = j\}, \tag{17}$$

where the sum on the right hand side is the number of rounds where $i$ is optimal but OptGTM pulls another arm (because it has reported a larger optimistic value).

Next, recall that $\theta^* \in C_{t,i}$ so that for $i_t^* = i$ it follows that

$$\text{UCB}_{t,i}(x_{t,i}) = \text{UCB}_{t,i}(x_{t,i}^*) \geq \langle \theta^*, x_{t,i}^* \rangle = \langle \theta^*, x_{t,i_t^*}^* \rangle.$$

Now, $i_t = j$ implies $\text{UCB}_{t,j}(x_{t,j}) \geq \text{UCB}_{t,i}(x_{t,i}) \geq \langle \theta^*, x_{t,i_t^*}^* \rangle$, since $i \in A_t$ for all $t$ and OptGTM selects the arm with maximal optimistic value. As a result, when $i_t^* = i$ and $i_t = j$, we obtain that

$$\text{UCB}_{t,j}(x_{t,j}) - \langle \theta^*, x_{t,j}^* \rangle \geq \langle \theta^*, x_{t,i_t^*}^* - x_{t,j}^* \rangle.$$

Importantly, we have shown in Lemma E.3 that the total difference on the left hand side is bounded before elimination, i.e., before round $\tau_j$. As a consequence, we get

$$
\begin{aligned}
\sum_{t=1}^{\tau_i} \mathbb{1}\{i_t^* = i, i_t = j\}\langle \theta^*, x_{t,i_t^*}^* - x_{t,j}^* \rangle &\leq \sum_{t=1}^{\tau_i} \mathbb{1}\{i_t^* = i, i_t = j\} \left( \text{UCB}_{t,j}(x_{t,j}) - \langle \theta^*, x_{t,j}^* \rangle \right) \\
&\leq \sum_{t=1}^{\tau_i} \mathbb{1}\{i_t = j\} \left( \text{UCB}_{t,j}(x_{t,j}) - \langle \theta^*, x_{t,j}^* \rangle \right) \\
&\leq 2\sum_{t=1}^{\tau_j} \mathbb{1}\{i_t = j\}\sqrt{\beta_{t,j}}\|x_{t,j}\|_{V_{t,j}^{-1}} + 4\sqrt{n_{\tau_j}(i)\log(T)} \\
&\leq \mathcal{O}\left(d\log(T)\sqrt{n_{\tau_j}(j)}\right) + 4\sqrt{n_{\tau_j}(i)\log(T)} \\
&\leq \mathcal{O}\left(d\log(T)\sqrt{n_{\tau_j}(j)}\right),
\end{aligned}
$$

where we first used Lemma E.3 and then Lemma E.4. Recalling that $\langle \theta^*, x_{t,i_t^*}^* - x_{t,j}^* \rangle > 0$ is constant for $j \neq i_t^*$ by assumption of a constant optimality gap, it then follows from Jensen's inequality that

$$\sum_{j \neq i}\sum_{t=1}^{\tau_i} \mathbb{1}\{i_t^* = i, i_t = j\} \leq \sum_{j \neq i}\mathcal{O}\left(d\log(T)\sqrt{n_{\tau_j}(j)}\right) \leq \mathcal{O}\left(d\log(T)\sqrt{KT}\right),$$

where we used that $\sum_{j \neq i} n_{\tau_j}(j) \leq T$. Hence, equation (17) yields

$$n_T(i) \geq n_T^*(i) - \mathcal{O}\left(d\log(T)\sqrt{KT}\right).$$

Since $\sigma_i$ must be a best response to $\sigma_{-i}$, we obtain

$$\mathbb{E}_{\boldsymbol{\sigma}}[n_T(i)] \geq \mathbb{E}_{\sigma_i^*, \sigma_{-i}}[n_T(i)] \geq n_T^*(i) - \mathcal{O}\left(d\log(T)\sqrt{KT}\right).$$

To obtain the result for $t \in [T]$l, suppose that on the good event $\mathcal{G}$ the contrary is true so that $n_t(i) < n_t^*(i) - \omega(d\log(T)\sqrt{KT})$. However, similarly to before, Lemma E.3 tells us that the number of rounds that are "poached" from arm $i$, i.e., $i_t^* = i$ and $i_t \neq i$, is upper bounded from above by $\mathcal{O}(d\log(T)\sqrt{Kt})$. Hence, since $\text{UCB}_{t,i}(x_{t,i}) \geq \langle \theta^*, x_{t,i_t}^* \rangle$ and $n_t(i) \leq n_t^*(i) - \omega(d\log(T)\sqrt{KT})$, it must hold that $\tau_i < t$. Then, on the good event $\mathcal{G}$, it follows that $n_{\tau_i}(i) \leq n_t(i) < n_t^*(i) - \omega(d\log(T)\sqrt{KT})$. Since event $\mathcal{G}$ has probability at least $1 - 1/T^2$, this implies

$$\mathbb{E}_{\boldsymbol{\sigma}}[n_T(i)] = \mathbb{E}_{\boldsymbol{\sigma}}[n_{\tau_i}(i)] \leq n_t^*(i) - \omega\left(d\log(T)\sqrt{KT}\right).$$

This contradicts the lower bound of $\mathbb{E}_{\boldsymbol{\sigma}}[n_T(i)] \geq n_T^*(i) - \mathcal{O}(d\log(T)\sqrt{KT})$.

$\square$

**Lemma E.8.** *Let $\boldsymbol{\sigma} \in \text{NE}(\text{OptGTM})$. Then,*

$$\mathbb{E}_{\boldsymbol{\sigma}}[n_T(i)] \leq \mathbb{E}_{\boldsymbol{\sigma}}[n_{\tau_i}^*(i)] - \mathcal{O}\left(d\log(T)K\sqrt{KT}\right),$$

*where the expectation on the right hand side is taken w.r.t. $\tau_i$.*

*Proof.* We have $\sum_{i=1}^{K} n_\tau^*(i) = \sum_{i=1}^{K} \sum_{t=1}^{\tau} \mathbb{1}\{i_t^* = i\} = \tau$ for any $\tau \in [T]$. Using Lemma E.7, we obtain

$$
\begin{aligned}
\mathbb{E}_{\boldsymbol{\sigma}}[n_T(i)] &= \mathbb{E}_{\boldsymbol{\sigma}}[n_{\tau_i}(i)] \\
&= \mathbb{E}_{\boldsymbol{\sigma}}\left[\sum_{t=1}^{\tau_i} \mathbb{1}\{i_t = i\}\right] \\
&= \mathbb{E}_{\boldsymbol{\sigma}}\left[\sum_{t=1}^{\tau_i} (1 - \mathbb{1}\{i_t \neq i\})\right] \\
&= \mathbb{E}_{\boldsymbol{\sigma}}[\tau_i] - \sum_{j \neq i} \mathbb{E}_{\boldsymbol{\sigma}}[n_{\tau_i}(j)] \\
&\leq \mathbb{E}_{\boldsymbol{\sigma}}[\tau_i] - \sum_{j \neq i} \mathbb{E}_{\boldsymbol{\sigma}}[n_{\tau_i}^*(j)] + \mathcal{O}\left(d\log(T)K\sqrt{KT}\right) \\
&= \mathbb{E}_{\boldsymbol{\sigma}}[n_{\tau_i}^*(i)] + \mathcal{O}\left(d\log(T)K\sqrt{KT}\right).
\end{aligned}
$$

$\square$

Combing the lower and upper bounds on each arm's utility of Lemma E.7 and Lemma E.8, we get

$$
n_T^*(i) - \mathcal{O}\left(d\log(T)\sqrt{KT}\right) \leq \mathbb{E}_{\boldsymbol{\sigma}}[n_T(i)] \leq \mathbb{E}_{\boldsymbol{\sigma}}[n_{\tau_i}^*(i)] - \mathcal{O}\left(d\log(T)K\sqrt{KT}\right). \tag{18}
$$

It then follows that

$$
\mathbb{E}_{\boldsymbol{\sigma}}[n_T^*(i) - n_{\tau_i}^*(i)] \leq \mathcal{O}\left(d\log(T)K\sqrt{KT}\right)
$$

so that

$$
\sum_{k=1}^{K} \mathbb{E}_{\boldsymbol{\sigma}}\left[n_T^*(i) - n_{\tau_i}^*(i)\right] \leq \mathcal{O}\left(d\log(T)K^2\sqrt{KT}\right).
$$

This concludes the proof of Lemma E.6. $\square$

Finally, recalling the regret decomposition from the beginning of the proof and using Lemma E.5 and Lemma E.6, we obtain for any $\boldsymbol{\sigma} \in \mathrm{NE}(\boldsymbol{\sigma})$ that

$$
R_T(\mathrm{GGTM}, \boldsymbol{\sigma}) \leq \mathcal{O}\left(\underbrace{d\log(T)\sqrt{KT}}_{\text{Lemma E.5}} + \underbrace{d\log(T)K^2\sqrt{KT}}_{\text{Lemma E.6}}\right) \leq \tilde{\mathcal{O}}\left(dK^2\sqrt{KT}\right).
$$

$\square$

### E.4 Linear Realizability of Reported Contexts

In the following, we comment on an interesting observation in the strategic linear contextual bandit that may also provide some insight into the effectiveness of OptGTM. Suppose that each arm reports its contexts in a linearly realizable fashion (without us explicitly incentivizing them to do so). Formally, we can express this as the following assumption.

**Assumption 3** (Linear Realizability of Reported Contexts). Every arm reports so that its reports follow some linear reward model. That is, for every arm $i \in [K]$, there exists a vector $\theta_i^* \in \mathbb{R}^d$ such that for all $t \in [T]$

$$
\langle \theta^*, x_{t,i}^* \rangle = \langle \theta_i^*, x_{t,i} \rangle. \tag{19}
$$

Perhaps surprisingly, the regret analysis of OptGTM becomes straightforward when the arms' strategies satisfy Assumption 3. Moreover, we can prove that OptGTM suffers $\tilde{\mathcal{O}}(d\sqrt{KT})$ strategic regret in every Nash equilibrium of the arms. That is, the regret guarantee is better than that of Theorem 5.2.

**A quick regret analysis.** Let $\boldsymbol{\sigma}$ be any NE under OptGTM. When we observe a reward $r_{t,i}$ after pulling arm $i$ in round $t$, we can interpret the reward as $r_{t,i} := \langle \theta^*, x^*_{t,i} \rangle + \eta_t = \langle \theta^*_i, x_{t,i} \rangle + \eta_t$. Hence, isolating arm $i$, the learner is essentially playing a linear contextual bandit with true unknown parameter $\theta^*_i$, contexts $x_{t,i}$, and rewards $r_{t,i} = \langle \theta^*_i, x_{t,i} \rangle + \eta_t$. As a result, the independent estimators $\hat{\theta}_{t,i}$ for every arm $i$, are in fact estimating $\theta^*_i$ and, according to Lemma E.1, with high probability $\theta^*_i \in C_{t,i}$. It is then also easy to see that OptGTM will never eliminate any of the arms with high probability. Now, since $\theta^*_i \in C_{t,i}$,

$$\langle \theta^*, x^*_{t,i} \rangle = \langle \theta^*_i, x_{t,i} \rangle \leq \mathrm{UCB}_{t,i}(x_{t,i}).$$

As a result, using Cauchy Schwarz inequality, we obtain

$$\mathrm{UCB}_{t,i}(x_{t,i}) - \langle \theta^*, x^*_{t,i} \rangle \leq \langle \hat{\theta}_{t,i} - \theta^*_i, x_{t,i} \rangle + \sqrt{\beta_{t,i}} \|x_{t,i}\|_{V^{-1}_{t,i}}$$

$$\leq \|\hat{\theta}_{t,i} - \theta^*_i\|_{V_{t,i}} \|x_{t,i}\|_{V^{-1}_{t,i}} + \sqrt{\beta_{t,i}} \|x_{t,i}\|_{V^{-1}_{t,i}} \leq 2\sqrt{\beta_{t,i}} \|x_{t,i}\|_{V^{-1}_{t,i}}.$$

In every round $t$, OptGTM selects the arm with maximal optimistic reported value $\mathrm{UCB}_{t,i}(x_{t,i})$ so that $\mathrm{UCB}_{t,i^*_t}(x_{t,i^*_t}) - \mathrm{UCB}_{t,i_t}(x_{t,i_t}) \leq 0$. We can then bound the instantaneous regret in round $t$ as

$$\langle \theta^*, x^*_{t,i^*_t} - x^*_{t,i_t} \rangle \leq \mathrm{UCB}_{t,i^*_t}(x_{t,i^*_t}) - \langle \theta^*, x^*_{t,i_t} \rangle$$

$$\leq \mathrm{UCB}_{t,i^*_t}(x_{t,i^*_t}) - \mathrm{UCB}_{t,i_t}(x_{t,i_t}) + 2\sqrt{\beta_{t,i_t}} \|x_{t,i_t}\|_{V^{-1}_{t,i_t}}$$

$$\leq 2\sqrt{\beta_{t,i_t}} \|x_{t,i_t}\|_{V^{-1}_{t,i_t}}.$$

Using the elliptical potential lemma and Jensen's inequality (Lemma E.4, [1, 24]), the total regret of OptGTM is given by

$$R_T(\mathrm{OptGTM}, \boldsymbol{\sigma}) \leq \sum_{i=1}^{K} \sum_{t \leq T \,:\, i_t = i} 2\sqrt{\beta_{t,i}} \|x_{t,i}\|_{V^{-1}_{t,i}} \leq \mathcal{O}\left(d \log(T)\sqrt{KT}\right).$$

We have thus shown the following guarantee.

**Corollary E.9.** *Suppose that Assumption 3 holds. Then,*

$$R_T(\mathrm{OptGTM}, \boldsymbol{\sigma}) = \tilde{\mathcal{O}}\left(d\sqrt{KT}\right)$$

*for every Nash equilibrium $\boldsymbol{\sigma} \in \mathrm{NE}(\mathrm{OptGTM})$.*

