# OpenReview forum: "Strategic Linear Contextual Bandits"
_NeurIPS.cc/2024/Conference — NeurIPS 2024 poster_

### Official Review · Reviewer_8u5z · 2024-07-07

**Soundness:** 4
**Presentation:** 3
**Contribution:** 3
**Rating:** 7
**Confidence:** 4

**Summary:**

The authors studied the problem of strategic agents who can modify their context in order to game the system under the linear contextual bandit framework. In this setting, each arm is a self-interested agent who wants to maximizes number of times it gets pulled by the learner. Prior work that did not explicitly consider the incentives of these arms suffers linear regret in the strategic setting. The authors proposed two mechanisms, first to deal with the case when the underlying parameter of the reward function is known to the learner in advance, and then when it is unknown to the learner. In either case, the proposed mechanism can incentivize the arms to be approximately truthful in reporting their true context to the learner. Furthermore, the authors provided strategic regret guarantee for each of the proposed mechanism that scales with $O(K^2 \sqrt{KT})$ when the latent parameter is known and $O(dK^2 \sqrt{KT})$ when the latent parameter is unknown.

**Strengths:**

- The studied problem of strategic linear contextual bandits is interesting, and the setup is novel.

- The paper is well-written and easy to follow.

- The theoretical bounds are clearly listed, and justifications for the differences in regret compared to non-strategic setting is provided.

**Weaknesses:**

- The paper did not provide matching lower bound analysis on the strategic regret in the two settings: when $\theta^*$ is known in advance and when it is unknown in advance by the learner.

- The notation in the Introduction is somewhat confusing. The authors refer to the strategic agents as arms, and use learner to denote the (what would typically be called) bandit agent. Furthermore, in the main contributions, the regret bound for the setting where $\theta^*$ is unknown in advance uses $d$, which has not been introduced at this point (and only introduced in Section 3 later on).

- The paper relied on a key assumption that the arms do not under-report their value. While this assumption seems intuitive, the authors did discussed in Appendix C that there might be cases where under-reporting may make sense for the arms.

- The authors did not provide experiments to support their theoretical findings.

**Questions:**

- Does manipulation budget for each arm matter in this setting? That is, does the current analysis change if the arms are only allowed to modify their true context by at most some (maybe unknown) values?

**Limitations:**

The authors have addressed the limitations of their work.

---

> ### Author Rebuttal · Authors · 2024-08-06
>
> Dear Reviewer 8u5z, thank you for taking the time to review our paper and your helpful comments. We respond to your questions and comments below.
>
> > - The paper did not provide matching lower bound analysis on the strategic regret in the two settings: when $\theta^*$ is known in advance and when it is unknown in advance by the learner.
>
> These are interesting questions for future work. In general, deriving lower bounds in strategic settings like ours is very challenging due to the intricate relationship between the learning algorithm and the equilibrium strategies that it induces. Note that we do inherit the $\Omega(d \sqrt{T})$ lower bound from the standard linear contextual bandit in the case where $\theta^*$ is unknown. However, as we conjecture in the discussion (Section 6), we believe the lower bound for the strategic problem to be $\Omega(d \sqrt{KT})$.
>
> > - The notation in the Introduction is somewhat confusing. The authors refer to the strategic agents as arms, and use learner to denote the (what would typically be called) bandit agent. Furthermore, in the main contributions, the regret bound for the setting where  $\theta^*$  is unknown in advance uses $d$, which has not been introduced at this point (and only introduced in Section 3 later on).
>
> Thanks for pointing this out. We made according modifications to the text addressing these things.
>
>
> > - The authors did not provide experiments to support their theoretical findings.
>
> Based on your and the other reviewers' suggestion, we added experiments to the paper. You can find the experiments and the experimental details in the rebuttal pdf.
>
> > - Does manipulation budget for each arm matter in this setting? That is, does the current analysis change if the arms are only allowed to modify their true context by at most some (maybe unknown) values?
>
> Thanks for the interesting question. Assuming a manipulation budget would not really change the setting. The main difference would be if we were to assume an extremely small manipulation budget of, e.g., constant size $C= O(1)$. For such small budgets, incentivizing the arms to be truthful is not necessary (after all, the amount of manipulation is almost insignificant). Instead, taking an adversarial (instead of strategic) approach would suffice, where we would slightly enlargen our confidence sets / increase exploration parameters to minimize regret. However, if the budget is large, then our approach, which does not rely on any assumption about the budget but instead bounds the effect of the arms' manipulation on our regret and the arms' utility, is necessary and much more effective. It is also worth mentioning that artificially assuming a manipulation budget can be unrealistic in practice and is one of the shortcomings of purely adversarial settings (see also reference [10] in the paper).

---

### Official Review · Reviewer_QjK5 · 2024-07-12

**Soundness:** 3
**Presentation:** 4
**Contribution:** 3
**Rating:** 7
**Confidence:** 3

**Summary:**

This paper studies a variant of the linear contextual bandit problem, where each arm is an agent and can strategically misreport its feature vector to the learner. The authors propose the Optimistic Grim Trigger Mechanism (OptGTM) that incentivizes the agents to report their feature vectors truthfully while simultaneously minimizing the regret. The paper first shows that if an algorithm does not explicitly consider the incentives of the agents, it can incur a linear regret. Then, the authors propose GGTM for known $\theta^*$ and OptGTM for unknown $\theta^*$, which achieve sublinear regrets.

**Strengths:**

1. The paper addresses an important and novel problem. The setting is innovative and has practical implications.
2. The algorithm design is simple and intuitive, and the theoretical analysis is rigorous.
3. The proposed OptGTM algorithm works for unknown $\theta^*$, which is an impressive result for me.
4. The readability is excellent (arguably the best among all the papers I reviewed). The problem setting is well-motivated, the model is clearly defined, the algorithm is easy to understand, and the flow of writing is logically coherent.

**Weaknesses:**

1. There is no experimental evaluation in the paper. While I understand that many theoretical papers do not include experiments, it would be beneficial to see some empirical results to validate the theoretical claims.
2. The assumption of the constant optimality gap (Lines 150-151) is unusual for regret minimization. Is there any reason for this assumption? It would be better if the authors could provide some rationale behind this assumption.

Update after rebuttal: The authors added some experiments.

**Questions:**

See Weaknesses.

**Limitations:**

Limitations are adequately discussed in the paper.

---

> ### Author Rebuttal · Authors · 2024-08-06
>
> Dear Reviewer QjK5, thank you for your time and your review. We respond to your comments below.
>
> > 1.  There is no experimental evaluation in the paper. While I understand that many theoretical papers do not include experiments, it would be beneficial to see some empirical results to validate the theoretical claims.
>
> Thanks for the suggestion. We added experiments to the paper, which you can find in the rebuttal pdf.  In the experiments, we adopt the perspective of each strategic arm greedily updating its strategy to maximize its utility. The results support our theoretical results and the effectiveness and generality of the proposed mechanism design approach.
>
> > 2.  The assumption of the constant optimality gap (Lines 150-151) is unusual for regret minimization. Is there any reason for this assumption? It would be better if the authors could provide some rationale behind this assumption.
>
> Without this assumption, the equilibrium behavior of the arms becomes difficult to analyze. For instance, consider the case where every time arm 1 is optimal it is only better than arm 2 by an amount $1/T$ (in terms of rewards). Then, roughly speaking, arm 2 only has to manipulate its contexts by a total amount $\leq 1 = T\times  1/T$ to "poach" all selections from arm 1. This tiny amount of manipulation by arm 2 is statistically undetectable so that we *cannot* prevent arm 1 losing all its utility when being truthful. In other words, we cannot promise arm 1 that truthfulness is a viable strategy (not much worse than untruthful reporting). This does not consequently mean that arm 1 will heavily manipulate its contexts in NE, but it does mean that we cannot use the truthful strategy as a "benchmark" when analyzing the NE (in the proofs of Theorem 4.2 and 5.2, we use that each arm's NE strategy must be a better response than the truthful strategy). We added a foonote to the main text explaining this and discuss this in an additional paragraph in the appendix.

---

> > ### Comment · Reviewer_QjK5 · 2024-08-10
> >
> > Thank you for your responses and the added experiments. I appreciate the effort, and my concerns have been addressed.

---

### Official Review · Reviewer_7kzE · 2024-07-13

**Soundness:** 3
**Presentation:** 3
**Contribution:** 3
**Rating:** 6
**Confidence:** 4

**Summary:**

This paper studies the strategic linear contextual bandit problem, where the agents can strategically change (report) their covariate to the principal. The authors propose an Optimistic Grim Trigger Mechanism (OptGTM) to encourage agents be truthful and achieve sublinear regret.

**Strengths:**

The authors design a new framework and new theoretical guarantees to both achieve the Nash equilibrium and minimize regret.

**Weaknesses:**

see my questions.

**Questions:**

1. The authors assumed that the inequality in Assumptions 1 and 2 holds. Could the authors change it to inequality after taking expectations over X? What if these assumptions fail?

2. Some related works are missing. Typically, when agents are strategic and incentivized to change their features, their private types can confound the observed covariates $x_{t,i}$ and the noise $eta_i$. Consequently, applying the loss in equation (3) would result in a biased estimator for $\theta$. Relevant works include:

Harris et al. Strategic Instrumental Variable Regression: Recovering Causal Relationships From Strategic Responses.
Yu et al.  Strategic Decision-Making in the Presence of Information Asymmetry: Provably Efficient RL with Algorithmic Instruments.

Could the authors also comment on this point?

3. Could the authors conduct some numerical studies to support their theoretical findings?

---

> ### Author Rebuttal · Authors · 2024-08-06
>
> Dear Reviewer 7kzE, thank you for reviewing our paper. Your time is highly appreciated. We respond to your questions below.
>
> > 1.  The authors assumed that the inequality in Assumptions 1 and 2 holds. Could the authors change it to inequality after taking expectations over X? What if these assumptions fail?
>
> We assume a possibly adversarial sequence of *true* contexts $x_{t,i}^*$. However, if we were to assume stochastically sampled contexts (which is a stronger assumption), we could indeed take an expectation over this distribution and our results still hold. When the assumption is violated and the arms can under-report their value, the problem appears to be intractable in some special cases (without other arguably stronger assumptions). We provided an example and discussion of this observation in Appendix C.
>
> >  2. Some related works are missing. Typically, when agents are strategic and incentivized to change their features, their private types can confound the observed covariates  $x_{t,i}$  and the noise etai. Consequently, applying the loss in equation (3) would result in a biased estimator for $\theta$. Relevant works include: Harris et al. Strategic Instrumental Variable Regression: Recovering Causal Relationships From Strategic Responses. Yu et al. Strategic Decision-Making in the Presence of Information Asymmetry: Provably Efficient RL with Algorithmic Instruments.
>
> Thank you for pointing us to these papers. While this is a quite different perspective on strategic learning to ours, we agree that strategic regression and, in general, feature confounding based on types is quite relevant to our work. We added a brief discussion highlighting similarities and differences to Section 2 (Related Work) under 'Strategic Learning'.
>
> > 3.  Could the authors conduct some numerical studies to support their theoretical findings?
>
> Following your and the other reviewers' suggestion, we have added experiments to the paper (see rebuttal.pdf). We believe thes experimental results are quite interesting and nicely illustrate the effectiveness of OptGTM and the necessity of a mechanism design approach in the strategic linear contextual bandit.

---

### Official Review · Reviewer_XxQv · 2024-07-17

**Soundness:** 3
**Presentation:** 4
**Contribution:** 3
**Rating:** 7
**Confidence:** 3

**Summary:**

The paper introduces a new strategic variant to the stochastic contextual bandit, where the contexts of the arms are not public but are made available to the arms only. The arms may choose to strategize by misreporting their context to sway the decisions of the learning algorithms. The model is analyzed in two settings: known and unknown \theta^*. The primary results are algorithmic (truthful mechanism) contributions and corresponding regret bounds.

**Strengths:**

The paper is exceptionally well-written and easy to understand and follow. The proofs are laid out well.

The strategic variant of linear contextual bandit is very neat and interesting. I look forward to more results, hopefully with fewer assumptions or in more general settings.

**Weaknesses:**

1) Regarding model description, my biggest concern is the assumption that the arms respond to the learning algorithm M in Nash Equilibrium. Specifically, the definition of NE for arm $i$ depends on future stochasticity (randomness in M and sampling noise or rewards) via \ eta_ T(i) . This is quite weird. What does it mean for a real-life setting? To calculate NE, each arm should not only know \theta^* but also the true contexts of other arms (in addition to future stochasticity). Is my understanding correct? I urge the authors to better explain this assumption (maybe in the appendix) with a real-life example and highlight the necessary information model.


2)In the result of theorem 4.2. Is it possible to give regret bound in terms of d, not K ? From my understanding of bandit literature, the cost of manipulation may be calculated in terms of d. Can you comment if it is possible to calculate the cost of the mechanism in terms of d instead of K?

**Questions:**

Please refer to the above comments and reply accordingly.

I am willing to engage in the rebuttal phase.

---

> ### Author Rebuttal · Authors · 2024-08-06
>
> Dear Reviewer XxQv, thank you for taking the time to read and review our paper. We respond to your questions below.
>
> > 1. Regarding model description, my biggest concern is the assumption that the arms respond to the learning algorithm $M$ in Nash Equilibrium. Specifically, the definition of NE for arm  i  depends on future stochasticity (randomness in M and sampling noise or rewards) via $\eta$. This is quite weird. What does it mean for a real-life setting? To calculate NE, each arm should not only know $\theta^*$ but also the true contexts of other arms (in addition to future stochasticity). Is my understanding correct? I urge the authors to better explain this assumption (maybe in the appendix) with a real-life example and highlight the necessary information model.
>
> Yes, you are correct in that the arms account for future stochasticity and contexts (in expectation). The reason for these assumptions is that we must ensure that the NE is well-defined. Without such knowledge, it is not clear what a best response for an arm is and we would instead need to, e.g., assume that the arms have a prior over the contexts etc. Even then, it can be unrealistic to assume that the arms reach an equilibrium (even in less complex models than ours). To this end, analyzing $\varepsilon$-NE can be helpful.
>
> In the added experiments (see rebuttal.pdf), we also study the case where the arms optimize their strategy over time using gradient ascent, which is a fairly natural model of strategic adaptation. Importantly, in this case, we do not need to assume that the arms have any prior knowledge (neither $\theta^*$ nor anything else). Instead, the arms learn to adapt their strategies purely based on sequential interaction. In future work, it would be interesting to also theoretically analyze these types of situations (e.g., arms are no-regret learners). We additional explanations to the main text and added a thorough discussion of this to the appendix. Thank you for bringing this to our attention. In the appendix, we now also mention that the regret bounds of Theorem 4.2 and 5.2 directly extend to the case where the arms do not reach equilibrium but instead play any $O(\sqrt{KT})$-NE and $O(d\sqrt{KT})$-NE, respectively.
>
>
> > 2. In the result of Theorem 4.2. Is it possible to give regret bound in terms of $d$, not $K$? From my understanding of bandit literature, the cost of manipulation may be calculated in terms of $d$. Can you comment if it is possible to calculate the cost of the mechanism in terms of $d$ instead of $K$?
>
> In the setting of Theorem 4.2, the latent parameter $\theta^*$ is known to the learner in advance. As a result, the regret bound does not involve $d$. In general, any dependence on the dimension $d$ is only expected when we have to learn $\theta^*$. Unfortunately, we also cannot artificially swap the dependence on $K$ for a dependence on $d$, as the reason for the $K$ dependence is the fact that we need to incentivize *all* strategic arms to be approximately truthful, which automatically yields a dependence on the number of arms.
>
> We are not really sure in what part of the literature the cost of manipulation is calculated in terms of $d$. In the literature on linear contextual bandits with adversarial corruptions, the manipulation is typically captured by an additive or multiplicative term $C$. Sometimes $C$ is defined as the total amount of context manipulation so that it is somewhat related to $d$ (however, $C$ is not expressed as a function of $d$).  Please let us know if you have any other questions or if we misunderstood your question.

---

> > ### Comment · Reviewer_XxQv · 2024-08-11
> >
> > Thank you for your detailed response. Your rebuttal indeed clarifies my concerns.
> > My recommendation (for any updated version) is that the authors include the discussion around the assumption of arms responding in N.E.

---

### Author Rebuttal · Authors · 2024-08-06

Thank you for taking the time to review our paper and for your helpful comments.

Reviewer 7kzE, Reviewer QjK5 and Reviewer 8u5z suggested to include experiments in the paper. Following your suggestions, we conducted simulations of strategic context manipulation, which we added to the paper (see rebuttal.pdf).

In the experiments, we study the situation where the strategic arms gradually adapt to the deployed learning algorithm to maximize their utility. Specifically, the arms repeatedly interact with the deployed bandit algorithm, updating their strategy (i.e., what contexts to report) at the end of each interaction using (approximate) gradient ascent w.r.t. their utility. In other words, the arms learn to maximize their utility in response to the deployed algorithm. This experimental setup should serve as a natural and simple model of real-world gaming behavior, which does *not* require any prior knowledge from the arms, and the experiments offer some additional insight into the effectiveness of OptGTM as well as the shortcomings of incentive-unaware algorithms such as LinUCB.

---

### Decision · Program_Chairs · 2024-09-25

**Decision:**

Accept (poster)

**Comment:**

This work studies the problem of Strategic Linear Contextual Bandits, where each of the arms could misreport their contexts with the goal of maximizing the number of times they are selected. The authors design a novel algorithm in the setting where the linear bandit parameter $\theta^*$ is unknown and show that the algorithm enjoys nice regret guarantees. The proposed problem is interesting from a theory perspective and might have some practical implications.

Most reviewers agree that the paper is very well written and that the presentation is excellent. Further all reviewers agree that the new setting is interesting and that the contributions are good. Some concerns that reviews have expressed are about how realistic it is to assume that arms are able to compute (approximate) NE (this likely requires knowledge of the algorithm $M$ and how other arms play),  optimality of the regret bounds, and lack of empirical evaluation. Authors address these concerns to the satisfaction of the reviewers and include some empirical evaluation of their algorithm.

Overall I find this paper interesting and with good theoretical contributions. I recommend that the authors include the response to the above reviewers concern in some way to the final version of this paper. Further, the authors should include their empirical evaluation as a new section to the paper. I am happy to recommend the paper for acceptance to the venue.